# Learning Polynomial Problems with $SL(2,\mathbb{R})$-Equivariance

**Hannah Lawrence**[*] **& Mitchell Tong Harris**[*]
Massachusetts Institute of Technology

## Abstract

Optimizing and certifying the positivity of polynomials are fundamental primitives across mathematics and engineering applications, from dynamical systems to operations research. However, solving these problems in practice requires large semidefinite programs, with poor scaling in dimension and degree. In this work, we demonstrate for the first time that neural networks can effectively solve such problems in a data-driven fashion, achieving tenfold speedups while retaining high accuracy. Moreover, we observe that these polynomial learning problems are equivariant to the non-compact group $SL(2,\mathbb{R})$, which consists of area-preserving linear transformations. We therefore adapt our learning pipelines to accommodate this structure, including data augmentation, a new $SL(2,\mathbb{R})$-equivariant architecture, and an architecture equivariant with respect to its maximal compact subgroup, $SO(2,\mathbb{R})$. Surprisingly, the most successful approaches in practice do not enforce equivariance to the entire group, which we prove arises from an unusual lack of architecture universality for $SL(2,\mathbb{R})$ in particular. A consequence of this result, which is of independent interest, is that there exists an equivariant function for which there is no sequence of equivariant polynomials multiplied by arbitrary invariants that approximates the original function. This is a rare example of a symmetric problem where data augmentation outperforms a fully equivariant architecture, and provides interesting lessons in both theory and practice for other problems with non-compact symmetries.

## 1 Introduction

Machine learning has emerged as a powerful tool for accelerating classical but time-consuming algorithms in scientific domains. In PDE time-stepping and molecular dynamics, for instance, productive lines of work have arisen around learning to forward-step in time or regress unknown parameters (Batzner et al., 2022; Han et al., 2018). These problems are solvable by existing methods, but can be significantly sped-up by training a neural net on a dataset of interest. Machine learning is especially valuable for problems for which, once a solution has been found, its correctness can be verified efficiently; this is the case for e.g. modeling catalysts, where a candidate output conformation can be confirmed with a few DFT steps (Chanussot et al., 2021).

Inspired by this line of research, we hypothesize that machine learning may allow for learning from specialized datasets of *polynomials*. Polynomials are ubiquitous in mathematics and engineering, and primitives such as minimizing a polynomial (Parrilo & Sturmfels, 2003; Jiang, 2013; Passy & Wilde, 1967; Nie, 2013; Shor, 1998) or certifying its positivity (Lasserre, 2001; Prestel & Delzell, 2013; Parrilo, 2000b), have attracted attention due to their direct applications to diverse problems ranging from operations research to control theory (Henrion & Garulli, 2005; Tedrake et al., 2010). Examples include certifying the stability of a dynamical system with a Lyapunov function, robot path planning, and designing nonlinear controllers (Ahmadi & Majumdar, 2016). Although there exist classical methods (ApS, 2022) for solving these problems, they are often computationally expensive.

A particularly attractive polynomial task for machine learning, which we discuss in further detail in Section 3, is predicting a matrix certificate of nonnegativity. This certificate can be efficiently checked independently of the neural network, and if the matrix is positive semidefinite (PSD), it *guarantees*

---

[*]Author order determined by coin flip.

that the original polynomial is nonnegative. Thus, one does not need to trust the trained network's accuracy; the utility of a certificate, once produced by any means, can be verified efficiently.

A trained neural network could not hope to outperform an existing semidefinite programming (SDP)-based technique on all instances; rather, we assume that there is an application-specific *distribution* of inputs (polynomials), and try to learn the shared attributes of such instances that may make a more efficient solution possible. This is similar to the assumption made when searching for sum of squares (SOS) certificates for nonnegativity: in general, it is NP hard to certify that a polynomial is nonnegative (Murty & Kabadi, 1985), but one could still search for a SOS certificate of low degree via SDPs, as we discuss in Section 3. One simple example of a class with exploitable structure could be polynomials all of the form $p_i(x) = s(x)^2 + c_i$, for $s(x)$ a real-rooted polynomial and constants $c_i$. If consistently present in a dataset, this structure can be leveraged to quickly compute global optima. Instead of discovering such features by hand, we *learn* them with a neural solver — ideally more quickly than off-the-shelf algorithms for minimization, positivity certification, and beyond.

In other contexts of structured inputs, it is now common practice to adapt neural architectures to the invariances of the input and output data types (Cohen & Welling, 2016). For example, architectures for point clouds are usually agnostic to translation, rotation, and permutation (Thomas et al., 2018a; Anderson et al., 2019), imparting improved sample efficiency and generalization (Elesedy & Zaidi, 2021). How can we design learning pipelines which take into account the underlying symmetries of *polynomial* learning tasks? In particular, the majority of polynomial tasks transform predictably under any linear change of variables. For example, the minimizer of a polynomial tranforms equivariantly with respect to linear transformations of the input variables, while the minimum value itself is invariant (Figure 1). Instead of working with the entire general linear group, we restrict our attention to the subgroup $SL(2, \mathbb{R})$. The special linear group $SL(2, \mathbb{R})$ consists of $2 \times 2$ matrices with determinant 1, which are sometimes called "area-preserving". Although the group is non-compact, which entails many difficulties (e.g. there is no finite invariant measure over a non-compact group, precluding the "lifting and convolving" approach of Finzi et al. (2020); see also Appendix C.1), the irreducible representations and the Clebsch-Gordan decomposition (defined in Section 4.2) provide building blocks for an equivariant architecture (Bogatskiy et al., 2020). In the case of $SL(2, \mathbb{R})$, they are both well-understood and naturally relate to polynomials. As a result, we can define an architecture that achieves *exact* equivariance to area-preserving linear transformations — subject only to numerical error from finite precision, and not to Monte Carlo integral approximations as in most previous work on Lie group symmetries (Finzi et al., 2020; MacDonald et al., 2021). To the best of our knowledge, this is the *only* known architectural technique to achieve exact $SL(2, \mathbb{R})$-equivariance.

However, we make a surprising discovery: although this well-established method (taking tensor products and linear combinations of irreps) can approximate any equivariant *polynomial*, it cannot approximate any equivariant *function*. In other words, we prove that there exists an $SL(2, \mathbb{R})$-equivariant function for which there is no converging sequence of equivariant polynomials multiplied by arbitrary invariants (Corollary 1), a result which may be of independent interest. Moreover, the function (equation 1) may be the very function one wants to approximate in positivity verification applications. Because of this impossibility result, we empirically explore other equivariance-inspired techniques, which obtain promising experimental results: data augmentation over well-conditioned subsets of $SL(2, \mathbb{R})$, as well as an equivariant architecture with respect to the subgroup $SO(2, \mathbb{R})$.

## 1.1 OUR CONTRIBUTIONS

In this work, our main contributions are threefold. First, we apply machine learning methods to certain polynomial problems that interest the optimization and control communities: positivity verification and minimization. To the best of our knowledge, this is the first application of machine learning to these problems. Second, we implement the first exactly $SL(2, \mathbb{R})$-equivariant architecture, which is universal over equivariant polynomials. Third, and perhaps most surprisingly, we prove that there exists an $SL(2, \mathbb{R})$-equivariant function for which there is no converging sequence of $SL(2, \mathbb{R})$-equivariant polynomials. This result indicates that no tensor product-based architecture can attain universality over $SL(2, \mathbb{R})$-equivariant functions. We therefore compare the performance of other equivariance-inspired techniques that do not enforce strict equivariance to the entire symmetry group. By closely evaluating equivariance with respect to $SL(2, \mathbb{R})$ on a concrete, richly-understood set of application problems, we provide a cautionary note on the incorporation of non-compact symmetries, and an invitation to lean more heavily on data augmentation in such cases.

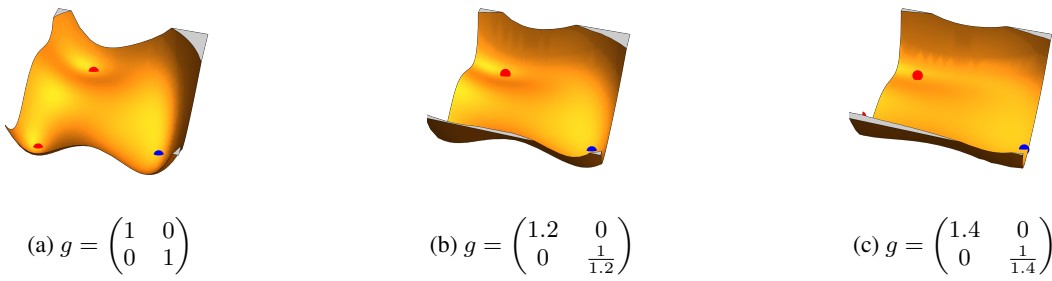

$$(a)\ g = \begin{pmatrix} 1 & 0 \\ 0 & 1 \end{pmatrix} \qquad (b)\ g = \begin{pmatrix} 1.2 & 0 \\ 0 & \frac{1}{1.2} \end{pmatrix} \qquad (c)\ g = \begin{pmatrix} 1.4 & 0 \\ 0 & \frac{1}{1.4} \end{pmatrix}$$

Figure 1: **The action of** $SL(2, \mathbb{R})$ **on polynomials.** Each plot is the result of applying some $g \in SL(2, \mathbb{R})$ to the degree 6 bivariate polynomial $\left((x-3)^2 + (y-3)^2\right)\left((x+2)^2 + \frac{1}{4}(y-2)^2\right)\left(\frac{1}{2}(x-2)^2 + (y+2)^2\right) - 8\left(x^2 + y^2\right)$, stretching or compressing the polynomial along each coordinate axis. The local minima are marked in red, and the global minimum in blue. While the minimum value is invariant, the minimizer transforms equivariantly under $SL(2, \mathbb{R})$.

## 1.2 RELATED WORK

**Equivariant Learning** Architectures enforcing equivariance to *compact* Lie groups abound in both theory and practice, often focusing on $SO(3)$ as it arises widely in practice. To name just a few, Cohen et al. (2018) enforces spherical equivariance using real-space nonlinearities, while Kondor et al. (2018) uses the tensor product nonlinearity. Several architectures designed for rotation-equivariant point cloud inputs also use tensor product nonlinearities (Thomas et al., 2018a; Anderson et al., 2019). Methods for more general Lie groups based on Monte Carlo approximations of convolution integrals include Finzi et al. (2020) and MacDonald et al. (2021), while Finzi et al. (2021) presents a computational method for solving for an equivariant linear layer of an arbitrary Lie group. Most relevant to our architecture is Bogatskiy et al. (2020), which introduces a Lorentz-group equivariant architecture closely related to ours; we discuss the relationship in more detail in Section 4, but note that our application area (and therefore the output layer), empirical findings, and universality properties are distinct. Finally, Gerken et al. (2022) provides one other empirical example where augmentation outperforms equivariance. However, they observe this phenomenon only for invariant tasks with respect to the compact group of rotations, whereas we work with a task equivariant to a non-compact group, and therefore hypothesize that the underlying mechanisms are quite different.

**Polynomial problems** Interest in polynomial problems is driven in part by the ubiquity of their applications. One application is to find a Lyapunov function to certify stability of a system, which can be accomplished with a semidefinite programming (SDP) solver (Parrilo, 2000a;b). SDP solvers are effective when the dimensions are not too large (Mittelmann, 2003). A different method to prove polynomial nonnegativity on an interval is to write the polynomial in the Bernstein basis (Farouki, 2012; Harris & Parrilo, 2023), which has been used to solve problems of robust stability (Garloff, 2000; Garloff et al., 1997; Malan et al., 1992; Zettler & Garloff, 1998). When the interval is not compact, however, the results are sensitive to the particular nonlinear map of the original interval to $[0, 1]$. For global polynomial minimization, a comparison of abstract approaches, including sum of squares-based methods, are given in Parrilo & Sturmfels (2003). Methods such as grid refinement and cutting plane algorithms can be used for many applications (Hettich & Kortanek, 1993), but the results are not certifiably feasible. One such application is in designing filter banks for signal processing. Perfect reconstruction is a desirable property of a filter bank, and is known to be equivalent to nonnegativity of a univariate trigonometric polynomial (Kortanek & Moulin, 1998; Moulin et al., 1997).

## 2 PRELIMINARIES

**Special Linear Group** Recall that a group $G$ is a set closed under an associative binary operation, for which an identity element $e$ and inverses exist. The group $SL(2, \mathbb{R})$ consists of $2 \times 2$ real-valued matrices of determinant 1. The condition number of a matrix $M$ is the ratio of its maximum to its minimum singluar values. (Note that $SL(2, \mathbb{R})$ contains arbitrarily poorly-conditioned matrices, as

e.g. diagonal matrices with entries $a$ and $\frac{1}{a}$ have condition number $a^2$.) $G$ acts on a set $\mathcal{X}$ if for every $g, h \in G$ and $x \in \mathcal{X}$, $g(hx) = (gh)x$ and $ex = x$. In this case, the orbit of $x_0 \in \mathcal{X}$ is $\{gx_0 | g \in G\}$. A homogeneous polynomial, or *form*, is one in which every term has the same degree. We are concerned with homogeneous polynomials in two real variables, or *binary forms*. Let $V(d)$ denote the vector space of binary forms of degree $d$, which can be identified with $\mathbb{R}^{d+1}$. For an element $p \in V(d)$, we may write it as $p$ or $p(\vec{x})$. The action of $g \in SL(2, \mathbb{R}) \subset \mathbb{R}^{2 \times 2}$ on a binary form $p$ is defined in Section 4.1 and shown in Figure 1.

**Representation Theory** A *representation* of a group $G$ is a vector space $V$ together with a map $\rho : G \to GL(V)$ satisfying $\rho(g_1)\rho(g_2) = \rho(g_1 g_2)$ for all $g_1, g_2 \in G$. An *irreducible representation*, or irrep, is a representation for which there does not exist a subspace $W \subseteq V$ satisfying $\rho(g)w \in W$ for all $g \in G, w \in W$. The irreps are crucial tools for understanding a group's other representations. The tensor product of representations $(V_1, \rho_1)$ and $(V_2, \rho_2)$ is given by $\rho : G \to GL(V_1 \otimes V_2), \rho(g) = \rho_1(g) \otimes \rho_2(g)$, and is itself a representation. A reducible representation $V_3$ may be decomposed into a direct sum of irreps $V_1$ and $V_2$. When $V_3$ is itself a tensor product of irreps, this is known as the Clebsch-Gordan decomposition. The relation $V_3 \simeq V_1 \oplus V_2$ means that there exists an invertible, equivariant, linear map $T : V_3 \to V_1 \oplus V_2$. An *equivariant* map with respect to a group $G$ is a map $f : V_1 \to V_2$ between representations $(V_1, \rho_1)$ and $(V_2, \rho_2)$ satisfying $f(\rho_1(g)v_1) = \rho_2(g)f(v_1)$ $\forall\, g \in G, v_1 \in V_1$.

## 3 TASK: SUM OF SQUARES CERTIFICATE OF POLYNOMIAL NONNEGATIVITY

For the first time, we propose machine learning for learning properties of a polynomial input. Many questions about polynomials, as discussed in Section 1, are invariant under a change of coordinates given by $SL(2, \mathbb{R})$. In this section, we introduce a widely applicable example, positivity verification, and we defer polynomial minimization to Appendix B. [1]

We discuss the following problem in depth because even though the problem can be solved with convex optimization, classical methods are slow for even moderately high degree or dimension polynomials, so an efficient learned solution could be useful in practice. Moreover, many problems about polynomials can be reduced to certifying nonnegativity (e.g. proving $\alpha$ is a lower bound on $p$ is equivalent to showing $p - \alpha \geq 0$).

One method to prove that a polynomial is nonnegative is by the semidefinite programming (SDP) method described in Parrilo (2000b); Lasserre (2001). Define the $d-$lift as $\vec{x}^{[d]} = \begin{pmatrix} y^d & xy^{d-1} & \cdots & x^d \end{pmatrix}^T$. Suppose $p(\vec{x}) = \vec{x}^{[d]^T} Q \vec{x}^{[d]}$ for some positive semidefinite $(d+1) \times (d+1)$ symmetric matrix $Q$. $\vec{x}^{[d]^T} Q \vec{x}^{[d]}$ is nonnegative by definition, so the equality to $p(\vec{x})$ implies $p$ is nonnegative for all $x$ and $y$. The problem of finding such a $Q$ is traditionally relegated to a convex optimization interior point solver. One convenient formulation given to a solver is

$$f(p) = \operatorname{argmax} \log \det Q \text{ such that } p(\vec{x}) = \vec{x}^{[d]^T} Q \vec{x}^{[d]} \text{ and } Q \succeq 0, \tag{1}$$

because this objective finds the analytic center (Boyd & Vandenberghe, 2004, §8.5) of the feasible region, which conveniently avoids the boundary of the feasible region. If $p$ is positive on $\mathbb{R}^2 \setminus \{(0, 0)\}$, then $f$ is well-defined as the optimization problem has a unique maximizer. The analytic center is a good choice precisely because of its scale-invariance and the following equivariance property.

For $A \in SL(2, \mathbb{R})$, define the *induced* matrix $A^{[d]}$ as the $(d+1) \times (d+1)$ matrix for which $(A\vec{x})^{[d]} = A^{[d]}\vec{x}^{[d]}$, as discussed in Marcus & Minc (1992); Marcus (1973); Parrilo & Jadbabaie (2008). The induced matrices describe how the irreps act; therefore, they are analogous to the Wigner d-matrices for $SO(3)$. Viewing a polynomial as an inner product between a basis and a coefficient vector, we get the important relation that $(A\vec{x})^T p = \vec{x}^{[d]^T}(A^{[d]^T} p)$. Therefore, the optimizer $f(p)$ has the equivariance property $f(p(A\vec{x})) = A^{[d]^T} f(p(\vec{x})) A^{[d]}$.

This application is well-suited for machine learning because we often only care if there exists a positive semidefinite $Q$. If the model predicts some matrix satisfying $p(\vec{x}) = \vec{x}^{[d]^T} Q \vec{x}^{[d]}$ and $Q \succeq 0$, then $p$ is automatically nonnegative; it is simple to validate this certificate of nonnegativity. This

---

[1]While this problem makes sense in higher dimensions (with $SL(d, \mathbb{R})$), we restrict to $d = 2$ for simplicity and defer higher dimensions to future work.

feature is unusual for machine learning applications, but highly advantageous. Even in safety-critical systems, a pipeline could include a neural network whose output is then certified.

# 4 $SL(2, \mathbb{R})$-EQUIVARIANT ARCHITECTURE

The equivariance property of $f$ in equation 1 suggests using an equivariant learning technique. A fully equivariant architecture for $SL(2, \mathbb{R})$ has not been previously presented. Indeed, since $SL(2, \mathbb{R})$ is a noncompact group, forming invariants by group averaging is impossible, as there is no finite invariant measure over $SL(2, \mathbb{R})$(see Appendix C.1 for details). However, there is nonetheless a straightforward strategy for equivariance. We adopt the architecture for noncompact groups given by Bogatskiy et al. (2020), which follows the established framework of taking tensor products between irreps and then applying the Clebsch-Gordan transformation (Thomas et al., 2018b; Kondor et al., 2018). Our fully equivariant $SL(2, \mathbb{R})$ architecture can be viewed as a natural specialization of their insightful Lorentz group architecture. In fact, the exposition simplifies dramatically because our binary form inputs lie precisely in the group's finite-dimensional irreps. Moreover, the Clebsch-Gordan decomposition is given by the classical transvectant, which has been well-studied in the invariant theory literature (see Section 4.1).

In Section 4.1, we provide the necessary background on the representation theory of $SL(2, \mathbb{R})$. In Section 4.2, we describe the $SL(2, \mathbb{R})$-equivariant architecture. In Section 4.3, however, we prove that **universality over equivariant polynomials does not provide universality over arbitrary smooth equivariant functions for $SL(2, \mathbb{R})$**. This is a surprising and general impossibility result, which contrasts with Bogatskiy et al. (2020) and casts doubt on the possibility of any universal $SL(2, \mathbb{R})$-equivariant architecture.

## 4.1 REPRESENTATION THEORY OF $SL(2, \mathbb{R})$

For the purposes of this paper, we are interested in finite-dimensional representations of $SL(2, \mathbb{R})$. For every positive integer $d$, the $(d + 1)$-dimensional vector space of degree $d$ binary forms ($V(d)$), is an irreducible representation of $SL(2, \mathbb{R})$. Throughout, we assume a monomial basis for $V(d)$ (i.e. $x^d$, $x^{d-1}y, \ldots, y^d$) and represent elements of $V(d)$ as vectors of coefficients in $\mathbb{R}^{d+1}$. Let $p \in V(d)$, and define the corresponding action on $V(d)$ by $\rho_d(g)p(\vec{x}) = p(g^{-1}\vec{x})$. These polynomial representations are all of $SL(2, \mathbb{R})$'s finite-dimensional irreps. Polynomial inputs are particularly well-suited to $SL(2, \mathbb{R})$-equivariance because they lie in the finite-dimensional irreducible representations.

Given two of these finite dimensional representations, $V(d_1)$ and $V(d_2)$, the tensor product representation decomposes as $V(d_1) \otimes V(d_2) \simeq \bigoplus_{n=0}^{\min(d_1, d_2)} V(d_1 + d_2 - 2n)$. The linear map verifying this isomorphism is an invertible, equivariant map we call $T$, which is the Clebsch-Gordan decomposition for $SL(2, \mathbb{R})$ (e.g. (Böhning & Bothmer, 2010)). A linear map is fully specified when defined on a basis, which we now do for $T$. Let $p(x_1, y_1) \in V(d_1)$ and $q(x_2, y_2) \in V(d_2)$. The $d_1 + d_2 - 2n$ component of $T(p \otimes q)$ is given by the classical $n$th order *transvectant* (see e.g. Olver (1999, §5)):

$$\psi_n(p, q) := \left( \left( \frac{\partial^2}{\partial x_1 \partial y_2} - \frac{\partial^2}{\partial x_2 \partial y_1} \right)^n p \cdot q \right) \Bigg|_{\substack{x=x_1=x_2 \\ y=y_1=y_2}} = \sum_{m=0}^n (-1)^m \binom{n}{m} \frac{\partial^n p}{\partial x^{n-m} \partial y^m} \frac{\partial^n q}{\partial x^m y^{n-m}} \tag{2}$$

where other authors, e.g. Böhning & Bothmer (2010), sometimes include a constant factor depending on the degrees and $n$. This is the isomorphism between the tensor product space and the ordinary spaces of binary forms.

## 4.2 ARCHITECTURE DETAILS

Our equivariant architecture is described in Algorithm 1 and in Figure 3. Each layer can be described by the nonlinear part and the linear part, and we describe each below. The inputs are elements of the finite dimensional irrep spaces of $SL(2, \mathbb{R})$.

**Nonlinearity** Each input channel has a list of polynomials, which we represent with vectors in $\mathbb{R}^d$. We compute all pairwise tensor products and decompose these tensor products with $T$ from Section 4.1, which gives a list of polynomials. We additionally apply a multi-layer perceptron ("LearnedMLP" in Algorithm 1) to any degree 0 polynomials.

---

**Algorithm 1** $SL(2, \mathbb{R})$-equivariant architecture

---

**Input:** $p_{init} \in \mathbb{R}^{d+1}$ where $\mathbb{R}^{d+1} \cong V(d)$, $d_1, \ldots, d_L$
**Output:** $M \in S^{\frac{d}{2}+1}$

1: $p_d = p_{init}$; `inputDegrees` $= \{d\}$; `outputDegrees` $= \{\}$     $\triangleright$ Irrep bookkeeping
2: **for** layer $\ell = 0, \ldots, L-1$ **do**
3:    Init $Q_r = [\,]$ for all $r$
4:    **for** $i, j$ in `inputDegrees` **do**
5:      **for** $n = 0, \ldots, \min(i, j)$ **do**
6:        Append $\psi_n(p_i, p_j) \in \mathbb{R}^{i+j-2n}$ to $Q_{i+j-2n}$    $\triangleright$ Nonlinearity and CG Decomp.
7:        Append $i + j - 2n$ to `outputDegrees`
8:    $p_0 = \text{LearnedMLP}([Q_0, p_0])$     $\triangleright$ MLP on Invariants + Skip Connection
9:    **for** $i$ in `outputDegrees` $\backslash \{0\}$ **do**
10:      $p_i = \text{LearnedLinearCombination}(Q_i, p_i)$    $\triangleright$ Linear Layer + Skip Connection
11:    `inputDegrees=outputDegrees`; `outputDegrees={}`
12: Return $L(p_0, p_1, \ldots, p_{d-1}, p_{init})$, as defined below    $\triangleright$ Final Linear Layer

---

**Learned linear layer** After the Clebsch-Gordan (CG) decomposition $T$, we have a collection of polynomials ranging from degree $0$ to degree $2d$. Gather all resulting polynomials of degree $k$ (from any tensor product and across any channel) into the vector of polynomials $v$. Let $\#k$ be the number of such polynomials. Let $c$ be the number of output channels. $W$ is a $c \times \#k$ learnable matrix. Then $Wv$ are the inputs of degree $k$ for the next layer. This operation is denoted by "LearnedLinearCombination" in Algorithm 1, where we additionally append to $v$ the input to the layer of degree $k$ as a skip connection.

**Last layer (general)** The inputs to the last layer $L$ are elements of the irreducible representation spaces, and the output lies in the vector space corresponding to some finite-dimensional representation. In other words, $L : \bigoplus_{n=0}^{d} V(n) \to V$ for some reducible representation $\rho$ with associated vector space $V$. Informally, Schur's Lemma describes the set of equivariant linear maps $L$ with this property as those which map linearly from the input irrep space to only matching irreps in $V$; see e.g. Stiefel & Fässler (2012, §2) for details. Although this methodology on its own is quite standard, in the next paragraph, we show a convenient way of choosing such an $L$ for positivity verification.

**Last layer (positivity verification)** Suppose (for this paragraph) the input polynomial is degree $2d$. The output space is $V = S^{d+1}$, symmetric $(d+1) \times (d+1)$ matrices, which decomposes into irreps as $\bigoplus_{k=0}^{d} V(2k)$ (the odd irreps do not contribute to symmetric matrices). An equivariant, linear map $L$ from $\bigoplus_{k=0}^{d} V(2k)$ to $S^{d+1}$ (which is unique up to a linear combination of channels by a real version of Schur's Lemma (Behboodi et al., 2022, §8)) can be conveniently precomputed using the transvectant; see Section A.1. For our application, we need $p(\vec{x}) = \vec{x}^{[d]^T} Q \vec{x}^{[d]}$. Although this may seem difficult to enforce, there is an elegant solution: we overwrite the degree $2d$ irrep with the input polynomial before computing $L$. This yields the desired property, because it can be shown (see Lemma 3) that $x^{[d]^T} L(0, \ldots, p_i, \ldots, 0) x^{[d]} = 0$ for any $i < 2d$, and $x^{[d]^T} L(0, \ldots, 0, p_{2d}) x^{[d]} = p_{2d}$.

This architecture can represent any equivariant polynomial in the following sense, which is a restatement of Lemma D.1 of Bogatskiy et al. (2020) (and a common primitive towards universality).

**Lemma 1.** *Let $p : V \to U$ be a polynomial that is equivariant with respect to $SL(2, \mathbb{R})$, where $V$ and $U$ are finite-dimensional representations of $SL(2, \mathbb{R})$. Then there exists a fixed choice of architectural parameters such that Algorithm 1 exactly computes $p$.*

## 4.3 Lack of universality

In this section, we prove that the architecture in Section 4.2 is not universal, in spite of Lemma 1. In particular, there are no parameter settings (representing an equivariant polynomial times an arbitrary invariant) that uniformly approximate the equivariant function $f$ described in equation 1.

**Theorem 1.** *Let $\mathcal{N}_W(p)$ denote the output of Algorithm 1 with (learned) parameters $W$ applied to the input $p$. Let $f$ be the continuous, $SL(2, \mathbb{R})$-equivariant function defined in equation 1. There exists*

*an input polynomial $p$, and an absolute constant $\epsilon > 0$ such that for any $W$, $|f(p) - \mathcal{N}_W(p)| > \epsilon$. Therefore, the architecture of Section 4.2 is not universal.*

This demonstrates more broadly that the function $f$ is not approximable by any sequence of equivariant polynomials (independent of a particular neural architecture). We highlight that this particular function is not an unnatural one dreamt up for the sake of this proof; rather, it is the very function we seek to approximate for positivity verification applications. We show that when $p = x^8 + y^8$, there is an algebraic obstruction to predicting $f(p)$ (see also Figure 4).

**Definition 1.** *Call the monomial $x^k y^r$ balanced mod $b$ if $k \equiv r \pmod{b}$. The polynomial $p$ is balanced mod $b$ if it is the weighted sum of monomials that are all balanced mod $b$.*

**Lemma 2.** *If $f$ and $g$ are balanced mod $b$, then for every $n$, $\psi_n(f, g)$ is balanced mod $b$.*

*Proof.* Suppose $f = \sum_i f_i$ and $g = \sum_j g_j$, where all $f_i$ and $g_j$ are monomoials balanced mod $b$. By bilinearity, $\psi_n(f, g) = \sum_{i,j} \psi_n(f_i, g_j)$. Therefore, it is enough to prove that $\psi_n(f_i, g_j)$ is the sum of balanced monomials. Let $f_i = x^{k_i} y^{r_i}$ and $g_j = x^{k_j} y^{r_j}$. A single term in the second definition of $\psi_n$ in equation 2 gives a monomial proportional to

$$\frac{\partial^n (x^{k_i} y^{r_i})}{\partial x^{n-m} \partial y^m} \frac{\partial^n (x^{k_j} y^{r_j})}{\partial x^m \partial y^{n-m}} \propto x^{k_i + k_j - n} y^{r_i + r_j - n}. \tag{3}$$

Since $k_i \equiv r_i$ and $k_j \equiv r_j$, then $k_i + k_j - n \equiv r_i + r_j - n \pmod{b}$. □

*Proof of Theorem 1.* Since $p = x^8 + y^8$ is balanced mod 8, every polynomial output from every layer will be as well by Lemma 2. (Note that neither the MLP applied to the invariants nor linear combinations of existing monomials will introduce new monomials.) Therefore, the final degree 4 output is proportional to $x^2 y^2$, implying there is no $x^4$ or $y^4$ term. For the $L$ described in Section 4.2, any linear combination of $L(x^8)$, $L(y^8)$, $L(x^4 y^4)$, $L(x^2 y^2)$, and $L(1)$ has a sparsity pattern of

$$\begin{pmatrix} \times & 0 & 0 & 0 & \times \\ 0 & 0 & 0 & \times & 0 \\ 0 & 0 & \times & 0 & 0 \\ 0 & \times & 0 & 0 & 0 \\ \times & 0 & 0 & 0 & \times \end{pmatrix} ; \text{ however, } f(p) \approx \begin{pmatrix} 1 & 0 & \boxed{-1.563} & 0 & 1/3 \\ 0 & \boxed{3.126} & 0 & -8/3 & 0 \\ \boxed{-1.563} & 0 & 14/3 & 0 & \boxed{-1.563} \\ 0 & -8/3 & 0 & \boxed{3.126} & 0 \\ 1/3 & 0 & \boxed{-1.563} & 0 & 1 \end{pmatrix} ,$$

which is a rounded/truncated result from ApS (2022). The sparsity pattern can be checked via the transvectant definition of $L$. Notice that the boxed numbers in the true solution (right) must be zero for any output of $\mathcal{N}_W(x^8 + y^8)$, regardless of $W$. Therefore, no output of the neural net can have the correct sparsity pattern. □

In light of Lemma 1, the problem is not inherent to our specific architecture, because our architecture can represent any equivariant polynomial. Instead, the problem must fundamentally be that $f$ cannot be approximated by equivariant polynomials. Therefore, we have the following corollary of potential independent interest, even outside of the machine learning community.

**Corollary 1.** *There exists a continuous, $SL(2, \mathbb{R})$-equivariant function and a compact subset of its domain for which no sequence of $SL(2, \mathbb{R})$-equivariant polynomials times $SL(2, \mathbb{R})$-invariant functions converge pointwise to the function on the subset.*

**What is different?** There are a few major differences between this setup and other related universality results. First, the function $f$ is only defined for a subset of binary forms. Another difference is the equivariance to the real group $SL(2, \mathbb{R})$ and not $SL(2, \mathbb{C})$. Theorem 4.1 of Bogatskiy et al. (2020) states that the analog of the architecture in Section 4.2 for $SL(2, \mathbb{C})$ is universal over equivariant functions. More generally, it is common to prove universality via polynomial approximation (Yarotsky, 2018; Dym & Maron, 2021). What goes wrong with such proofs in the real case? One problem is that the algebra of invariants with respect to the group, and the algebra of invariants with respect to a *maximally compact subgroup*, may differ for the real-valued case, as noted by Haddadin (2021).[2]

---

[2]For example, $SO(2, \mathbb{R})$ is the maximal compact subgroup of $SL(2, \mathbb{R})$ (Lang, 1985, p. 19). However, $a + c$ is an invariant of the polynomial $ax^2 + bxy + cy^2$ under an $SO(2, \mathbb{R})$ action, but is not an invariant under an $SL(2, \mathbb{R})$ action.

| Model Name | Description |
|---|---|
| mlp-aug-rots | MLP with rotation augmentations |
| mlp-aug-$a$-$b$ | MLP with augmentations by $g^{[d]}$ and $g$ has condition number $\kappa$ s.t. $a \leq \kappa \leq b$ |
| SO2Net-aug-$a$-$b$ | $SO(2, \mathbb{R})$ equivariant network with data augmented by $g^{[d]}$ as above |
| SO2Net | $SO(2, \mathbb{R})$ equivariant architecture with no data augmentation |
| SL2Net | $SL(2, \mathbb{R})$ equivariant architecture with no data augmentation |
| MLP | multilayer perceptron with no data augmentation |

Table 1: Model names and meanings

**Implications of these results.** Nearly all equivariant universality results of which we are aware rely on polynomial approximation as a crucial intermediate step (Yarotsky, 2018; Dym & Maron, 2021; Bogatskiy et al., 2020). Moreover, the exhibited hard function is exactly the one we wish to approximate in many applications. Therefore, it is unclear whether any $SL(2, \mathbb{R})$-equivariant architecture holds promise for this class of problems. Instead, we propose returning to *data augmentation*, as well as an architecture equivariant to $SL(2, \mathbb{R})$'s well-studied maximal compact subgroup, $SO(2, \mathbb{R})$. One might hope to use a clever combination of (non-polynomial) renormalization and $SO(2, \mathbb{R})$-equivariance to circumvent our impossibility result and recover full $SL(2, \mathbb{R})$-equivariance; however, we prove in Proposition 2 of Appendix A.2 that this is not possible. For the sake of completeness, we include the SL2Net architecture in our experiments, but find that it underperforms alternatives.

## 5 EXPERIMENTS

Since learning on polynomial input data is a novel contribution, there do not yet exist standardized benchmarks. Therefore, we design our own tasks: polynomial minimization and positivity verification. We have released all data generation (as well as training) code, so that future research may build on these preliminary benchmarks. We defer an exploration of polynomial minimization to Appendix B.

Our experiments are divided based on the underlying distribution of polynomials used to generate synthetic data, described in further detail in Appendix B.2. The second distribution is a structured class of polynomials that are optimal for the Delsarte linear programming bound for the cardinality of a spherical code (Delsarte et al., 1977), providing a realistic distribution of "natural" polynomials. An interior point method (ApS, 2022) is used to generate the values of $f(p)$ in equation 1.

There are several practical considerations when augmenting by elements of $SL(2, \mathbb{R})$. First, since $SL(2, \mathbb{R})$ is not a compact group, *any data augmentation induces a distribution shift* (see Appendix C for a proof). This is in contrast to compact groups, where any datapoint in an orbit is just as likely as any other datapoint in that orbit. Therefore, $SL(2, \mathbb{R})$-equivariant methods aid primarily in out-of-distribution generalization, rather than in-distribution sample complexity. Further challenges associated with the non-compactness of $SL(2, \mathbb{R})$ are discussed in Appendix C.

In our experiments, we compare several instantiations of equivariant learning. The first is the $SL(2, \mathbb{R})$ equivariant architecture described in Section 4.2. The second is a class of multilayer perceptrons (MLPs). We train the MLPs on an *augmented* training distribution: starting with the same training set, we apply random $SL(2, \mathbb{R})$ transformations of the training polynomials. We control the conditioning of the augmentation matrices both to avoid numerical issues and to induce more or less dramatic distribution shifts to obtain several different trained MLPs. The third architecture is an $SO(2, \mathbb{R})$ equivariant model, whose details are described in Appendix A.3. We train additional versions of the $SO(2, \mathbb{R})$ equivariant architecture by including data augmentations of various condition numbers. A description of the model names are given in Table 1. Experiment details for all experiments, as well as equivariance tests for the various models, are included in Appendix B.

**Timing Comparison: Trained Network vs Solver** An impetus for turning to machine learning was to find faster ways of solving polynomial problems in practice. We compare the run time of a first order SDP solver with the wall time of a trained MLP on a CPU in Table 2. Since Mosek (ApS, 2022) was used to generate the data distribution, those values are used as ground truth. The MLP is about two orders of magnitude faster, while still very accurate, as reported in Table 2. While the true test will be for much higher degrees, where traditional SDP solvers become prohibitive, these results are strong experimental motivation to pursue machine learning for this task.

| Degree | 6 | 8 | 10 | 12 | 14 |
|---|---|---|---|---|---|
| MLP NMSE | 9.5e-6 | 6.0e-5 | 2.9e-5 | 2.3e-5 | 1.1e-5 |
| MLP times (min) | 0.062 | 0.082 | 0.17 | 0.22 | 0.29 |
| SCS NMSE | 2.7e-5 | 6.2e-5 | 1.2e-4 | 2.7e-4 | 1.2e-3 |
| SCS times (min) | 3.50 | 4.94 | 9.11 | 18.8 | 37.4 |

Table 2: Comparison of the MLP to the first order solver SCS (O'Donoghue et al., 2016). The mean squared errors are with respect to the ground truth labels computed by the second order solver, Mosek (ApS, 2022). The times are estimates for 5,000 test examples on a single CPU.

**Comparison of Equivariance and Augmentation for Out-of-Distribution Generalization** In Figure 2, we report the test errors for each of the models. We did not only test on the test dataset; we also randomly augmented the test dataset with $A \in SL(2, \mathbb{R})$ transformations. The average value of the condition number of $A^{[d]}$ for the transformations on the test set is labeled on the horizontal axis. The leftmost points are the test errors on the original test dataset. The rightmost points are the test errors resulting from transforming the test dataset with transformations with large condition numbers.

**Observations** On the untransformed test set, the best results are the MLP trained with augmentations close to rotations, followed by the SO2Net and unaugmented MLP. An advantage of the SO2Net is that it has about half as many parameters (see Appendix B.2 for model parameter counts). Using high condition numbers for the *training* augmentation of an MLP increases the test error of the untransformed dataset, validating our assertions in Appendix C.1 about distribution shifts induced by poorly conditioned matrices. Similarly, some models have this behavior when they encounter distribution shifts in the *test* set: the MLP, MLP augmented with rotations, and SO2Nets share the trend that the test error increases as the test distribution undergoes more transformations.

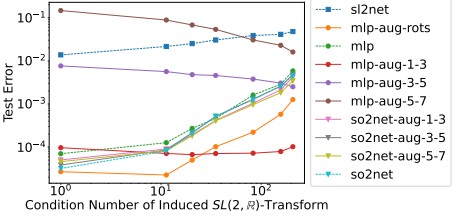

(a) Test errors for max det on random polynomials

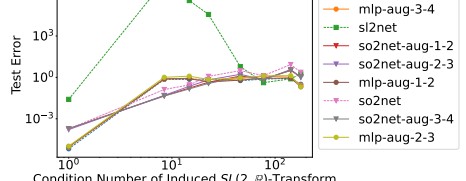

(b) Test errors for max det on polynomials arising from spherical code bounds

Figure 2: **Test errors.** The test dataset is transformed by random $A \in SL(2, \mathbb{R})$. The average value of the condition numbers of $A^{[d]}$ for the transformations is labeled on the horizontal axis. The computed loss is the MSE normalized by the ground-truth labels' norm (from the SDP) and with an additive term to avoid dividing by zero, $\frac{\|\mathcal{N}(p) - f(p)\|_{\text{Fro}}^2}{1 + \|f(p)\|_{\text{Fro}}^2}$, and averaged over 3 independent runs.

## 6 CONCLUSION

In this work, we demonstrated the promise of machine learning for polynomial problems, particularly with equivariant learning techniques. Even a simple MLP can vastly outperform an SDP solver in terms of speed, while equivariance-based approaches such as data augmentation and rotation-equivariance can improve out-of-distribution generalization. Surprisingly, although we construct an exactly $SL(2, \mathbb{R})$-equivariant architecture capable of representing any $SL(2, \mathbb{R})$-equivariant polynomial (demonstrating that one can overcome some of the superficial difficulties of equivariance to noncompact groups), it turns out that this does not suffice for universality. Therefore, any approach to equivariant architectures based on tensor products or polynomial approximation likely fails.

As future work, it would be interesting to consider paradigms for equivariance that sidestep the fundamental roadblock of polynomial approximation. One promising direction is frame averaging (Puny et al., 2021) for $SL(2, \mathbb{R})$. From the perspective of applications, our proof-of-concept experiments provide a strong signal to pursue the acceleration of polynomial optimization with machine learning.

ACKNOWLEDGMENTS

We thank Pablo Parrilo for inspiring the project and many productive conversations. HL is supported by the Fannie and John Hertz Foundation and the NSF Graduate Fellowship under Grant No. 1745302. MH is supported by the NSF Graduate Fellowhip under Grant No. 2141064.

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

TABLE OF CONTENTS

## A   TECHNICAL BACKGROUND AND DETAILS

### A.1   FURTHER ARCHITECTURE DETAILS

In Figure 3, we provide a visual of the $SL(2,\mathbb{R})$-equivariant architecture.

We now describe how we compute the linear equivariant map (known as an intertwiner) $L$.

We first construct the linear, equivariant map $L^{-1}$ from $V$ to irreps. Once we have defined $L^{-1}$, it is trivial to obtain $L$ via a one-time matrix inversion. To obtain $L^{-1}$, we will define $L^{-1}$ on each basis element of $V$, given by $e_i \otimes e_j + e_j \otimes e_i$ where $e_i$ represents the monomial $x^i y^{2d-i}$. Therefore, $L^{-1}(M)$ is in fact the Clebsch-Gordan decomposition of the element $M$ in the tensor product representation space, since it describes the change of basis from tensor products of irreps to individual irreps. As we have seen, the transvectant from equation 2 is precisely this equivariant map between the spaces $V(d) \otimes V(d)$ and $\bigoplus_{k=0}^{d} V(2k)$. Therefore, $L^{-1}(e_i \otimes e_j + e_j \otimes e_i) = \{\psi_n(e_i, e_j)\}_{n=0}^{2d}$ is the inverse of the last layer. One can verify that this $L^{-1}$ is linear and has the appropriate equivariance property.

The following lemma implies that setting the degree $2d$ component to the original polynomial $p$ in the last layer correctly enforces the constraint $x^{[d]^T} M x^{[d]} = p(x)$.

**Lemma 3.** *If $L$ is the last layer as described above, $x^{[d]^T} L(0, 0, \ldots, p) x^{[d]} = p(x)$ and $x^{[d]^T} L(0, \ldots, 0, p, 0, \ldots 0) x^{[d]} = 0$.*

*Proof.* The last layer is linear and equivariant by construction. First, we note that every binary form is the sum or difference of $2d$th powers of linear forms. (Each form in the sum/difference has 2 free parameters, so with an arbitrary number we can write any binary form). Therefore, it is enough that

$$x^{[d]^T} L(0, 0, \ldots, x^{2d}) x^{[d]} = x^{[d]^T} \begin{pmatrix} 0 & 0 & \cdots & 0 \\ 0 & 0 & \cdots & 0 \\ \vdots & \vdots & \ddots & \vdots \\ 0 & 0 & \cdots & 1 \end{pmatrix} x^{[d]^T} = x^{2d}, \qquad (4)$$

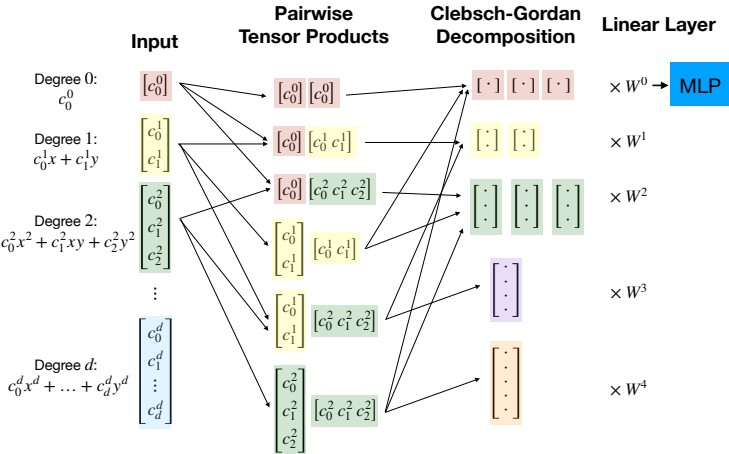

Figure 3: **Our $SL(2,\mathbb{R})$-equivariant layer.** We use color to indicate in which irrep's vector space a given activation resides, i.e. the degree of polynomials in that vector space. The inputs to the layer live in the finite-dimensional irreps of $SL(2,\mathbb{R})$, corresponding to homogeneous polynomials of different degrees. The nonlinear layer takes all pairwise tensor products of these input polynomials, and the resultant matrices are decomposed back into elements of the irreps using the Clebsch-Gordan decomposition of $SL(2,\mathbb{R})$. A simple consequence of Schur's Lemma is that the only possible equivariant linear layer is one that linearly combines elements within each irrep's vector space individually, which is reflected in our linear layer. (Any isomorphism from an irrep to itself can be written as linear combinations of independent channels from the same irrep according to a real version of Schur's Lemma (Behboodi et al., 2022, §8).) Finally, we apply an MLP to the invariants. The outputs of the layer again reside in the irrep vector spaces, and so we can compose several layers of this form.

where the second equality follows because $x^d \otimes x^d$ represents the matrix and $\psi_n(x^d, x^d) = 0$ for all $n > 0$. Therefore, the map given by $p \mapsto x^{[d]^T} L(0, 0, \ldots, p) x^{[d]}$ is an invertible linear transformation with rank $d + 1$. On the other hand $v \mapsto x^{[d]^T} L(v) x^{[d]}$ is a linear map from a space of dimension $(d + 1)(d + 2)/2$ to a space of dimension $d + 1$. By the rank-nullity theorem, the null space has dimension $(d + 1)(d + 2)/2 - (d + 1)$, which means that all remaining dimensions of the input are in the null space. $\qquad\square$

## A.2 Scale homogeneity and $SO(2,\mathbb{R})$ vs $SL(2,\mathbb{R})$ equivariance

In this work, we have evaluated both $SL(2,\mathbb{R})$ and $SO(2,\mathbb{R})$ equivariant architectures. Because its a bigger group, $SL(2,\mathbb{R})$ equivariance is naturally stronger than $O(2)$ equivariance. However, the polynomial positivity verification problem explored in Section 5 has the additional structure that it is not only equivariant, but also 1-scale homogeneous: if homogeneous polynomial $p(\vec{x})$ has verifier matrix $M$ of maximal determinant, then $cp(\vec{x})$ has verifier matrix $cM$ of maximal determinant. (Similarly, polynomial minimization is also 1-homogeneous, although it is $SL(2,\mathbb{R})$-*invariant*, not equivariant.) The first implication of this structural property is that we may enforce it via data normalization, as discussed below:

**Data normalization**  If a function $f$ is 1-homogeneous, and one has an approximating function $g$, it is possible to make $g$ 1-homogeneous via the following procedure:

$$\tilde{g}(x) := \|x\| g\left(\frac{x}{\|x\|}\right)$$

Indeed, this is the normalization we employ in all of our experiments, as data normalization (having all inputs roughly of the same scale) is a widespread and important technique to achieve good neural network performance during training. However, it is important to note that even if $g$ is

$SL(2,\mathbb{R})$-equivariant, $\tilde{g}$ may not be. If $A \in SL(2,\mathbb{R})$, then

$$\tilde{g}(Ax) = \|Ax\|g\Big(\frac{Ax}{\|Ax\|}\Big) \stackrel{?}{=} A \circ \tilde{g}(x) = A \circ \left[\|x\|g\Big(\frac{x}{\|x\|}\Big)\right] = \|x\|g\Big(\frac{Ax}{\|x\|}\Big)$$

However, $\|Ax\| \neq \|x\|$ in general; the uncertain equality does not hold unless $g$ is already 1-homogeneous.

One might still reasonably ask whether $SL(2,\mathbb{R})$ and $SO(2,\mathbb{R})$ equivariance are actually that different in the presence of scale equivariance, since scale equivariance implies that all interesting behavior in the function to be learned is captured by its behavior on the unit ball. In this section, we expound upon the distinctions between $SO(2,\mathbb{R})$ equivariance and $SL(2,\mathbb{R})$ equivariance for scale-equivariant functions. Importantly, for our equivariant problem of polynomial positivity verification, the two notions are distinct — this is proven in a following proposition. Therefore, although the normalization procedure we employ slightly breaks equivariance (when there is no normalization), capturing both properties at once is a promising future direction. Moreover, as shown in subsequent plots, our architecture still captures equivariance better than other architectures.

We begin with the following example, which is our only example in which the two notions coincide. However, as explained below, it is a trivial case.

**Proposition 1** ($O(n)$ and $SL(n,\mathbb{R})$ *invariance* are equivalent for scale-invariant problems)**.** *Let* $f : \mathbb{R}^n \to \mathbb{R}$ *be a scale-invariant function, i.e.* $f(cx) = cf(x)$*, and assume both* $O(n)$ *and* $SL(n,\mathbb{R})$ *act on* $\mathbb{R}^n$*. Then* $f$ *is* $SL(n,\mathbb{R})$*-invariant if and only if it is* $O(n)$*-invariant.*

*Proof.* The statement follows from the simple fact that a function that is both scale-invariant *and* $O(n)$-invariant must be constant. This is because $O(n)$-invariance implies that $f(x)$ depends only on $\|x\|$, while scale invariance implies that $f(x)$ depends only on $\frac{x}{\|x\|}$. $\qquad\square$

The above example is not very satisfying. In the next proposition, we prove that $SO(2,\mathbb{R})$ and $SL(2,\mathbb{R})$ equivariance are distinct for the polynomial positivity problem considered in Section 3 – even when considering 1-homogeneous functions via input normalization. The following proposition states that this is not equivalent to an $SO(2,\mathbb{R})$ transformation of $p$.

**Proposition 2** ($SO(2,\mathbb{R})$ *equivariance* is not equivalent to $SL(2,\mathbb{R})$ *equivariance* and scale homogeneity)**.** *There exist* $p$ *and there exists some element* $A \in SL(2,\mathbb{R})$ *such that there is no rotation* $R \in SO(2,\mathbb{R})$ *satisfying*

$$\frac{A^{[d]}p}{\|A^{[d]}p\|} = R^{[d]}p.$$

This proposition is somewhat counterintuitive. If $A$ is $n \times n$, then for every $x \in \mathbb{R}^n$, there exists an orthogonal matrix $R$ depending on $R$ and $x$ such that $\frac{Ax}{\|Ax\|} = Rx$. The reason this case is different is because even though induced matrices are size $(d+1) \times (d+1)$, they only have 4 degrees of freedom. This idea is formalized below.

*Proof.* Suppose for the sake of contradiction that for all $A \in SL(2,\mathbb{R})$, there is some rotation $R \in SO(2,\mathbb{R})$ and $\alpha \in \mathbb{R}$ satisfying

$$\frac{A^{[d]}p}{\|A^{[d]}p\|} = R^{[d]}p.$$

Left-multiplying both sides by $R^{[d]T}$, we obtain

$$R^{[d]T}\frac{A^{[d]}p}{\|A^{[d]}p\|} = R^{[d]T}R^{[d]}p = p.$$

Let $M^{[d]} = \frac{1}{\|A^{[d]}p\|}R^{T^{[d]}}A^{[d]}$. Such an $M$ exists because $(R^T A)^{[d]} = R^{T^{[d]}}A^{[d]}$ as the induced matrix map is a homomorphism, and the map is homogeneous. So the condition in the equation above is equivalent to $p(M^{-1}x) = p(x)$. For a generic $p$, it seems natural that there are only finitely many

$M$ such that $p(M^{-1}x) = p(x)$. In general, this is a system of $d+1$ degree $d$ polynomial equations with 4 variables. For this proposition, however, we only need to show this system has finitely many solutions for a particular $p$.

In particular, we argue that if $p = x^{2d} + y^{2d}$, then there are only finitely many $M$. The terms other than $x^{2d} + y^{2d}$ of $(ax + by)^{2d} + (ex + fy)^{2d}$ would need to vanish. A generic coefficient in the expansion is a constant times $a^i b^{2d-i} + e^{2d-i} f^i$. Any term with even $i$ and $2d - i$ would have no solution over the reals unless $ab = 0$ and $ef = 0$. The first and last expansion terms give us that $a^{2d} + e^{2d} = 1$ and $b^{2d} + f^{2d} = 1$. If for instance $a = b = 0$, then $e = \pm 1$ and $f = \pm 1$. There are only finitely many cases of which variables are zero to consider.

Next we argue that $MA^{-1}$ cannot be a multiple of an orthogonal matrix (which will be a contradiction to the equality $M^{[d]} = \frac{1}{\|A^{[d]}p\|} R^{T^{[d]}} A^{[d]}$). There exists some $A \in SL(2, \mathbb{R})$ such that $MA^{-1}$ is not a multiple of an orthogonal matrix for any of these finite number of $M$. The matrix $\begin{pmatrix} r & 0 \\ 0 & 1/r \end{pmatrix}$ is in $SL(2, \mathbb{R})$ for every $r \in \mathbb{R} \setminus \{0\}$. We will choose $A$ so that $A^{-1}$ is of this form. If $M = Q_M R_M$ is the QR decomposition of $M$, and $R_M = \begin{pmatrix} u & v \\ 0 & w \end{pmatrix}$, then choose $r$ so that $r^4 \neq \frac{v^2 + w^2}{u^2}$ for every $R_M$. Then the norms of $MA^{-1} \begin{pmatrix} 1 \\ 0 \end{pmatrix} = Q_M \begin{pmatrix} ru \\ 0 \end{pmatrix}$ and $MA^{-1} \begin{pmatrix} 0 \\ 1 \end{pmatrix} = Q_M \begin{pmatrix} v/r \\ w/r \end{pmatrix}$ are different. Since $Q_M$ is orthogonal, $MA^{-1}$ is not a multiple of an orthogonal matrix. $\qquad\square$

**Remark 1.** *As a result of the above proposition, $SO(2, \mathbb{R})$ and $SL(2, \mathbb{R})$ equivariance* are not *equivalent for positivity verification. This is because representations of $SL(2, \mathbb{R})$ capture a richer variety of transformations than induced matrices of $O(2)$, even after normalization.*

Although the proposition above proves that normalization and $SO(2, \mathbb{R})$-equivariance together still do not suffice to capture $SL(2, \mathbb{R})$-equivariance, the following reasoning still captures why we might expect $SO(2, \mathbb{R})$-equivariance to perform well. To approximate the $SL(2, \mathbb{R})$-equivariant, 1-homogeneous function $f$ on an input $x$, first normalize $x$, and then input it to an $SO(2, \mathbb{R})$-equivariant architecture, and finally re-scale by $\|x\|$. (More precisely, the $SO(2, \mathbb{R})$ input representation in the architecture is merely the subrepresentation of the original $SL(2, \mathbb{R})$ input representation.) Although, by the previous proposition, this technique is not $SL(2, \mathbb{R})$-equivariant, it still captures equivariance.

### A.3 $SO(2, \mathbb{R})$ EQUIVARIANT ARCHITECTURE

The main development for the SO2Net was making such a network compatible with the input and output types required for the tasks. The input for all tasks was a binary homogeneous polynomial of degree $d$.

**Input layer** The first layer maps the input to elements of the representation spaces of $SO(2, \mathbb{R})$. We map the homogeneous polynomial in $(x, y)$ to a homogeneous polynomial in $(\cos\theta, \sin\theta)$ and then compute its Fourier coefficients.

**Nonlinearity** In the case of $SO(2, \mathbb{R})$, the tensor product simplifies immensely. Because the elements of the representation spaces of degrees $j$ and $k$ are scalars, their tensor product is simply their product that lives in the representation space of order $j + k$.

**Learned linear layer** We are free to add arbitrary linear combinations of scalars that live in the same representation spaces. These weights are learned parameters. Furthermore, we learn an MLP to apply to each set of invariants (Fourier modes of order 0). We send the results of this layer into either another nonlinear layer or into the final layer.

**Final layer** The final layer is output dependent. In the case of an invariant problem, the final layer should return an element of the representation space of order 0. For the problem of Section 3, we convert these Fourier modes into a sequence of irreps of $SL(2, \mathbb{R})$ and apply the final layer used for the SL2Net. We convert a channel of Fourier modes to an ordinary binary form as follows. Every Fourier term of the form $a\cos(k\theta) + b\sin(k\theta)$ corresponds to a homogeneous polynomial in $\cos(\theta)$ and $\sin(\theta)$ of degree $k$. If $k < d$, where $d$ is the degree of the binary form that we need, we

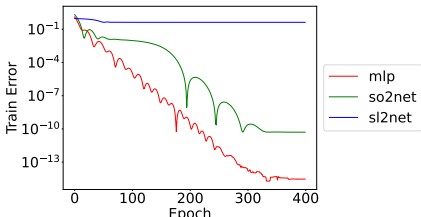

Figure 4: Training curves (MSE loss) for the single input datapoint $x^8 + y^8$, with maximum determinant matrix in equation 1 as the label to fit. As described in Section 4.3, the SL2Net cannot fit this example.

multiply by the necessary power of $1 = \cos^2(\theta) + \sin^2(\theta)$ to lift to a form of degree $d$. Then we reverse-substitute $(x, y)$ for $(\cos\theta, \sin\theta)$. This can be used as input for the last layer of the SL2Net.

The construction of this SO2Net satisfies the requirements of Theorem D.1 of Bogatskiy et al. (2020), which implies that any equivariant map for our tasks can be uniformly approximated by this network.

## B    EXPERIMENTS

### B.1    HARD EXAMPLE

Here, we validate in Figure 4 that the SL2Net cannot learn the hard function at the single input datapoint $x^8 + y^8$, whereas the SO2Net and MLP can.

### B.2    EXPERIMENTAL SETUP

All experiments were run on Nvidia Volta V100 GPUs, using the AdaM optimizer with learning rate $3 \cdot 10^{-4}$. Roughly, training on a single GPU took 15-30 minutes for each MLP, 1.5-2.5 hours for each SL2Net, and 2.5-6 hours for each SO2Net. For runs with data augmentation, augmentations were performed using $10{,}000$ presaved $2 \times 2$ $SL(2, \mathbb{R})$ matrices (and their induced versions) with condition numbers in the specified range. Details of the distribution from which these matrices were generated, as well as from which the random augmentations were applied in our test error plots, can be found in the code at github.com/harris-mit/polySL2equiv.

**Data distribution: Random, rotationally symmetric** Random positive degree $d$ forms were generated as follows. A real Wigner matrix $A$ of dimensions $(d+1) \times (d+1)$ is sampled. Each entry is an independently and identically distributed random normal variable with mean 0 and variance 1. Then $p = \vec{x}^{[d]^T}(A^T A + 10^{-8} I)\vec{x}^{[d]}$. The identity perturbation is added to ensure strict positivity. The maximum determinant Gram matrix was computed with ApS (2022).

**Data distribution: Delsarte spherical code bounds** Given a constant $\alpha \in [-1, 1]$ and dimension $d$, a spherical code is a set $X(\alpha) \subseteq \mathbb{R}^d$ of unit vectors such that for all $x, y \in X$, $\langle x, y \rangle \in [-1, \alpha]$. The Delsarte bound gives an upper bound for the cardinality of such a code. The best such bound (Delsarte et al., 1977) is a solution to the optimization problem

$$
\begin{aligned}
\min_{\{g_k\} \geq 0, g_0 = 1} \quad & \sum g_k G_k(1) \\
\text{s.t.} \quad & \sum g_k G_k \leq 0 \text{ on } [-1, \alpha],
\end{aligned}
\tag{5}
$$

where $G_k$ is the $k$th Gegenbauer polynomial (Szegő, 1939, §4.7) of weight $\frac{d-2}{2}$ scaled by $\frac{d+2\cdot k-2}{d-2}$. Therefore, we can define $G_k$ recursively as $G_0(x) = 1$, $G_1(x) = dx$, and

$$\lambda_{k+1} G_{k+1}(x) = x G_k(x) - (1 - \lambda_{k-1}) G_{k-1}(x).$$

If the optimal polynomial is $p(x) = \sum_k g_k G_k(x)$, we know that $q(x) = -(x^2 + 1)^d p(\frac{-x^2+\alpha}{x^2+1}) \geq 0$ for all $x \in \mathbb{R}$. For our experiments we fix $d = k_{\max} = 3$ and sample $\alpha$ uniformly at random in $[-1, 1]$. Each of these $\alpha$ gives another polynomial $q(x)$ after solving equation 5. We add $10^{-4}$ so they are strictly positive and then homogenize these results. Our distribution is the collection of such $q(x)$.

| Architecture | Hyperparameters | Parameters |
|---|---|---|
| MLP | hidden layers 100-1000 (dimension 100, then 1000) | 117,816 |
| SL2Net | 5 layers, 50 channels, max internal deg. 12, invariant MLP 10-10 | 887,771 |
| SO2Net | 3 layers, 10 channels, max internal deg. 12, invariant MLP 10-10 | 58,110 |

Table 3: Architecture hyperparameters for the max determinant experiment on random rotationally-symmetric data. For multi-layer perceptron (MLP) architectures, "$x$-$y$" indicates $x$ hidden units, followed by $y$ hidden units, etc.

| Architecture | Hyperparameters | Parameters |
|---|---|---|
| MLP | hidden layers 100-1000 (dimension 100, then 1000) | 117816 |
| SL2Net | 3 layers, 20 channels, max internal deg. 7, invariant MLP 100-1000 | 16021 |
| SO2Net | 3 layers, 20 channels, max internal deg. 7, invariant MLP 100-1000 | 76070 |

Table 4: Architecture hyperparameters for the max determinant experiment on Delsarte spherical code data. For multi-layer perceptron (MLP) architectures, "$x$-$y$" indicates $x$ hidden units, followed by $y$ hidden units, etc.

**Positivity Verification Setup** We used $5,000$ training examples, $500$ validation examples, and $500$ test examples. Experiments were trained for 700 epochs across 4 random seeds. We used the hyperparameters shown in Tables 3 and 4, which were chosen heuristically by comparing validation errors across a small number ($< 20$) of runs.

**Minimization Setup** For the purpose of computing the minimum of an inhomogeneous polynomial, the polynomial was generated as follows. Let $m$ be sampled from a standard normal distribution. Let $(x_0, y_0)$ each be the result of sampling independently from a standard normal and taking the absolute value. If $a, b$ are sampled independently from a uniform distribution on $[0, 1]$, then let $(x_1, y_1) = ((2a - 1)x_0, (2b - 1)y_0)$. Then multiply a collection of polynomials from the five quadratics in $\{((x \pm x_0)^2 + (y \pm y_0)^2), ((x - x_1)^2 + (y - y_1)^2)\}$ to get a degree $d - 2$ polynomial. Add together every product from this collection that has degree $d - 2$ to get the polynomial $p$ and then return $p \cdot ((x - x_1)^2 + (y - y_1)^2) + m$. The minimum of this polynomial is guaranteed to be $m$ and occur at $(x_1, y_1)$. After expanding the resulting polynomial, we can separate the polynomial into binary forms of degrees $0, \ldots, d$ and input all forms simultaneously into the first layer of the neural network. We used $5,000$ training examples, $100$ validation examples, and $100$ test examples.

Experiments were trained for 400 epochs across 2 random seeds. We used the hyperparameters shown in Table 5, which were chosen heuristically by comparing validation errors across a small number ($< 20$) of runs.

**Normalization** As noted in the previous section, we normalize input polynomials via $p \mapsto \frac{p}{\|p\|}$, where $\|p\|$ is equal to the $L_2$ norm of the polynomial's coefficients, in a monomial basis. Since our problems are 1-homogeneous, we rescale the outputs via $\|p\|$. This is done for all architectures we compare; therefore, all methods are scale equivariant (or in particular, 1-homogeneous) during training and validation.

### B.3 POLYNOMIAL MINIMIZATION

In this section, we repeat our experiments for an alternative $SL(2, \mathbb{R})$-equivariant problem: polynomial minimization. If a bivariate polynomial (not necessarily homogeneous) has a unique minimum,

| Architecture | Hyperparameters | Parameters |
|---|---|---|
| MLP | hidden layers 100-1000 (dimension 100, then 1000) | 104,901 |
| SL2Net | 3 layers, 20 channels, max internal deg. 12, invariant MLP 10-10 | 163,628 |
| SO2Net | 3 layers, 10 channels, max internal deg. 12, invariant MLP 10-10 | 48,930 |

Table 5: Architecture hyperparameters for the minimization experiment. For multi-layer perceptron (MLP) architectures, "$x$-$y$" indicates $x$ hidden units, followed by $y$ hidden units, etc.

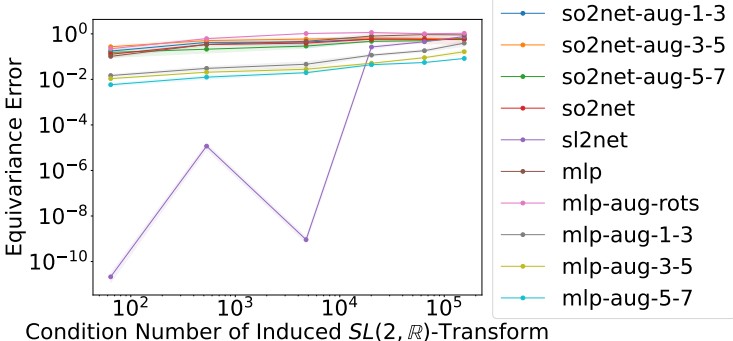

Figure 5: Equivariance errors for different architectures on the polynomial minimization problem, averaged over two random seeds with a standard-deviation error bar.

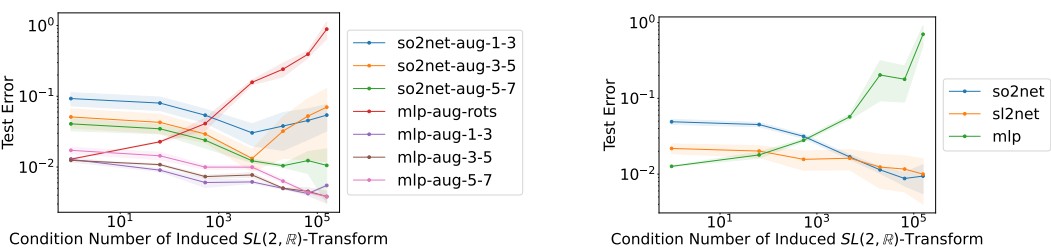

Figure 6: Normalized test errors for different architectures on the polynomial minimization problem, averaged over two random seeds with a standard-deviation error bar.

then that global minimum is invariant to any invertible change of coordinates, including $SL(2, \mathbb{R})$. The $SL(2, \mathbb{R})$-equivariant architecture is amenable to this problem by the following adaptations.

- The input to the first layer is a sequence of forms of different degrees. The degree $k$ form are all the terms of degree $k$ of the original polynomial.
- The last layer returns an element of the degree 0 representation space – an invariant scalar.

The changes to SO2Net are analogous:

- The input to the network is a sequence of forms of different degrees. We implement this as several input channels. We interpret each as a polynomial in $\cos(\theta)$ and $\sin(\theta)$ and compute the Fourier transform. We zero-pad the Fourier coefficients so that all channels have the same number of Fourier coefficients, and this collection of channels is the input to the first nonlinear layer.
- The output returns the 0th Fourier mode of the last layer, again an invariant scalar.

We again compare various architectures for applying machine learning to this problem. As shown in Figure 5, the $SL(2, \mathbb{R})$-equivariant architecture is far more equivariant than any other architecture, with or without augmentations. However, as shown in Figure 6, the so2 and sl2 equivariant architectures have an advantage over an MLP only for very ill-conditioned $SL(2, \mathbb{R})$ transformations. Moreover, MLPs with $SL(2, \mathbb{R})$-augmentations perform the best in terms of test error. The takeaway is similarly that the use-case should inform which model type is preferable.

### B.4    EQUIVARIANCE TESTS FOR POSITIVITY VERIFICATION

**Equivariance Error**    The equivariance error of the models in Table 1 after transformation of the input by random $A \in SL(2, \mathbb{R})$ with average condition number of $A^{[d]}$ given on the horizontal axis are reported in Figure 7. For $p$ a polynomial, $y$ the training label (maximum determinant matrix), and

$\mathcal{N}(p)$ the network's output on $p$, the equivariance error was calculated as

$$\frac{\|A^{[d']T}\mathcal{N}(p)A^{[d']} - \mathcal{N}(A^{[d]}p)\|}{\|A^{[d']T}yA^{[d']}\|},$$

averaged over all $p$ in the test set, where the norm is the Frobenius norm.

As shown, the SL2Net architecture was the most equivariant at every transformation level. Almost all of the MLPs also learned some level of equivariance.

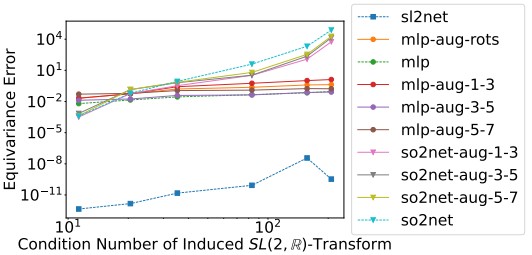
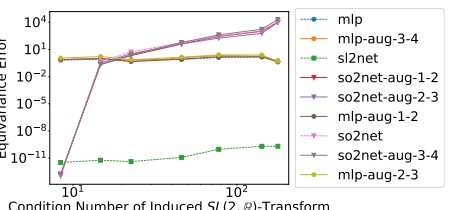

Figure 7: Equivariance error for positivity verification

**Impact of architecture**  In Figure 8, we also study the impact of the equivariant architecture hyperparameters on the equivariance error of the network. We find that the higher degree the internal irreps, the further the equivariance error is from numerical precision. In the following section, we hypothesize an explanation for this behavior.

## C    PRACTICAL CONSIDERATIONS FOR $SL(2, \mathbb{R})$

### C.1    DISTRIBUTION SHIFT

There are several crucial difficulties unique to non-compact groups, more fundamental even than the equivariant architectural design question. All of them arise due to the following essential fact: a non-compact group $G$ does not have a finite Haar measure. Recall that the Haar measure $\mu$ is a left-invariant measure over a compact group, i.e. for any subset $S \subseteq G$, $\mu(S) = \mu(gS) \ \forall g \in G$. Since a non-compact $G$ does not have a Haar measure integrating to 1, for any finite measure $\mu$ on $G$, there exists a subset $S \subseteq G$ and $g \in G$ such that $\mu(S) \neq \mu(gS)$. A similar statement can be made about a space $\mathcal{X}$ on which $G$ acts; for most group actions (and all those we consider), there exists a subset $S \subseteq \mathcal{X}$ and $g \in G$ such that $\mu(S) \neq \mu(gS)$.

What is the implication of this for machine learning? If $\mathcal{X}$ is our data space, then the data distribution is a distribution $p$ over $\mathcal{X}$. For compact groups, it is nearly always assumed in prior work that $p$ is invariant under $G$: in other words, any datapoint in an orbit is just as likely as any other datapoint in that same orbit. For example, a molecule in space is equally likely to appear in a dataset in any 3D orientation. When the data distribution is itself invariant, augmenting a data point $x$ and its label $y$ as $(gx, gy)$ (where $g$ is drawn according to the Haar measure) effectively generates a fresh sample from the data distribution, conditioned on the particular orbit. Both data augmentation and equivariance are therefore eminently reasonable from the perspective of sample complexity, as all points in a given orbit are equally likely (and, for $G$ a subgroup of the orthogonal group $O(n)$, equally "important": $\|gx\| = \|x\|$). Note that for the special case of the translation group, even though it is non-compact, it is trivial to emulate this "nice" behavior by centering the input data as a pre-processing step. This provides a simple example of frame-averaging (Puny et al., 2021), but an analog is not known for $SL(2, \mathbb{R})$.

In contrast, any non-trivial probability distribution over $\mathcal{X}$ is not the same as $g\mathcal{X}$ for $g \in G$ when $G$ is non-compact. **Therefore, data augmentation induces a distribution shift.** In this sense, non-compact equivariance is inextricably linked to the notion of "extrinsic equivariance," introduced by Wang et al. (2022), in which group transformations change the support of the data distribution. As noted in Wang et al. (2023), extrinsic equivariance can be helpful *or* harmful, and we still lack a good

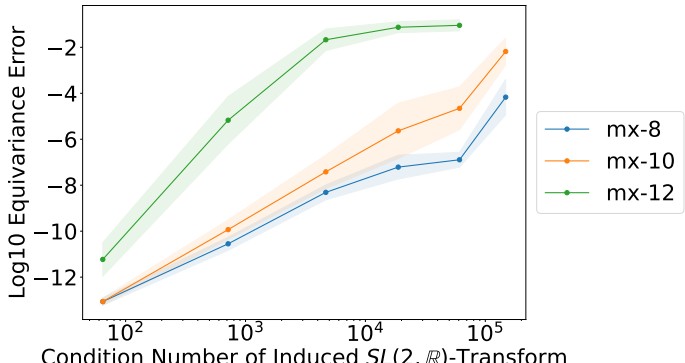

Figure 8: The impact of the $SL(2, \mathbb{R})$-equivariant architecture on equivariance error. Architectures with channels in {10, 30, 50}, layers in {3, 4, 5}, and maximum internal irrep degree stored in {8, 10, 12} (indicated by `mx-8`, `mx-10`, etc) are trained for a short time (30 epochs). The log equivariance error is then averaged across all architectures with the same maximum internal degree. As shown, architectures with a lower internal degree tend to be more equivariant. The variation in equivariance due to the other hyperparameters other than maximum irrep degree is minimal – changing the maximum irrep degree has the biggest effect on equivariance.

framework for understanding when either case will hold. In sum, we view $SL(2, \mathbb{R})$-equivariance as aiding in out-of-distribution generalization, rather than sample complexity, where higher condition numbers induce more dramatic distribution shifts. This is an important distinction, and one that will arise when we compare an $SO(2, \mathbb{R})$-equivariant architecture on an $SO(2, \mathbb{R})$-invariant distribution with a $SL(2, \mathbb{R})$-equivariant architecture.

## C.2    INDUCED CONDITION NUMBER STUDY

First, we demonstrate in Figure 9 the exponential relationship between the condition number (denoted by $\kappa$) of $A$ and that of $A^{[d]}$. Each line is a randomly generated matrix $A \in SL(2, \mathbb{R})$, and demonstrates a linear relationship between $d$ and $\log(\kappa(A^{[d]})$. This is likely a relationship that one can prove, but we defer such a theorem to future work.

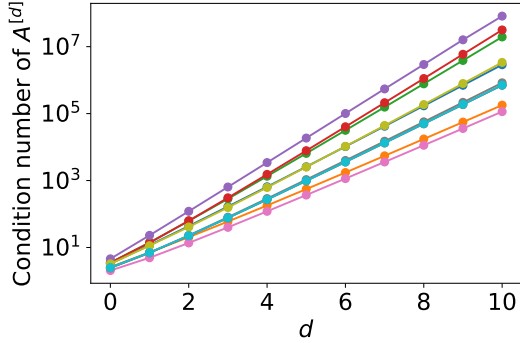

Figure 9: The condition number of $A^{[d]}$ as a function of $d$ for 10 randomly generated matrices $A \in SL(2, \mathbb{R})$. Here, each colored line corresponds to a different sampled matrix.

## C.3    CONDITION NUMBER IMPLICATIONS

Past work on noncompact Lie groups has largely focused on how to design linear equivariant layers, but not much on the practical difficulties that arise from working with a group of non-orthogonal matrices. Here, we expound upon some of these difficulties, in the hopes that it may be useful for future practitioners working with groups which are not subgroups of the orthogonal group.

One gnarly consequence of the non-unitarity of a given $SL(2,\mathbb{R})$ representation is that data norms vary wildly along an orbit: if $g \notin O(2)$, $\|gx\| \neq \|x\|$. This makes the choice of loss function important, and something one must bear in mind when assessing out of distribution generalization.

Unlike elements of an orthogonal group, the elements of $SL(2,\mathbb{R})$ can be arbitrarily poorly conditioned. As shown in the previous subsection, Appendix C.2, even if $A \in SL(2,\mathbb{R})$ has a moderate condition number (e.g. less than 10), $A^{[10]}$ may have a condition number several orders of magnitude larger. This affects various aspects of training and testing. First, when $A^{[d]}$ is poorly conditioned, it means that computed values of $p(Ax)$ and therefore $M(p(Ax))$ may be untrustworthy. This means that there is a limit to how much data augmentation can be done, and how reasonable it is to test on transformed datasets.

Our architecture is designed to be fully equivariant. This means that it should handle arbitrary transformations of the input data. When testing equivariance of our model, it is apparent that the errors from poorly conditioned transformations compound with more Clebsch-Gordan layers and higher irrep degrees. High degree polynomials in the monomial basis are poorly conditioned, and low degree polynomials may have numerical issues when we repeatedly multiply them together. This means there is a practical limitation with how big the networks can be before the internal computations become unstable. This is consistent with Figure 8 from the previous section: networks with higher degree internal irrep activations are slightly less equivariant than those with lower degrees, as the architecture — by nature of its very equivariance — is constrained to apply an ill-conditioned matrix to its internal activations, as a result of the representation matrix applied to the input.

One could hope to choose a different basis for each vector space of homogeneous polynomials, such that the resulting induced representation is better-conditioned. This is an open problem for future work, and indeed may not always be possible for the entire group, as shown in the following general theorem about non-unitary representations.

**Proposition 3.** *Let $G$ be a group, and $\rho : G \to GL(V)$ a representation of $G$ with associated $n$-dimensional vector space of $V$ and such that $O(n) \subseteq Im(\rho(g))$. A basis change $U \in O(n)$ of $V$ is a mapping[3] from $\rho$ to $\rho'$, where $\rho'(g)x = U\rho(g)U^Tx$. Then, there is no unitary basis change of $\rho$ under which $\max_{g \in G} \kappa(\rho(g))$ changes.*

*Proof.* We wish to find a basis for the vector space $V$ corresponding to a representation p such that $\max_{g \in G} \kappa(\rho(g))$ is as small as possible. A basis change here corresponds to the mapping $\rho(g) \mapsto U\rho(g)U^T$. Therefore, we want to solve: $\min_{U \in O(n)} \max_{g \in G} \kappa(U\rho(g)U^T)$. If $Im(\rho(g))$ contains $O(n)$, then for any $U \in O(n)$, we can define $h$ as the preimage of $U$: $\rho(h) = U$. Then, $U\rho(g)U^T = \rho(hgh^{-1})$, so $\max_{g \in G} \kappa(U\rho(g)U^T) = \max_{g \in G} \kappa(\rho(g))$ is independent of $U$. This implies that no basis is better than any other basis — at least from a worst-case perspective. $\square$

**Remark 2.** *The standard, two-dimensional (irreducible) representation of $SL(2,\mathbb{R})$ satisfies the criterion of Proposition 3, as $SL(2,\mathbb{R})$ contains $SO(2,\mathbb{R})$. Therefore, there is no basis under which this representation is better-conditioned – at least, in a worst-case sense over $SL(2,\mathbb{R})$.*

In spite of Proposition 3, there is still reason to believe a basis may exist that performs well in practice. First, the condition that $O(n) \subseteq Im(p(g))$ is restrictive; for representations like the induced representations $A^{[d]}$ of $G = SL(2,\mathbb{R})$ for $d > 2$, this does not hold. (Consider just that there are on the order of $n$ free parameters for elements in $O(n)$ but only 4 for elements of $SL(2,\mathbb{R})$.) Moreover, perhaps we do not care about the metric $\max_{g \in G} \kappa(p(g))$, but rather something distributional: $\mathbb{E}_{g \sim \mu} \kappa(p(g))$. In this case, we could hope to find a basis change that is better-conditioned on high-probability group elements.

Finally, one practical consideration to do with $SL(2,\mathbb{R})$-equivariance and conditioning is the loss function. Although the $L_2$ norm is unchanged under orthogonal transformations — so that loss functions are often themselves invariant – the $L_2$ norm is changed by non-unitary representations, and therefore not invariant under our non-unitary $SL(2,\mathbb{R})$ representations. As shown below, for the positivity verification problem, the normalized loss function we use in our experiments may be distorted by a factor of up to $\kappa^2$. This means that, when we query the loss of an equivariant model on an $SL(2,\mathbb{R})$-transformed datapoint, the loss may vary in accordance with the condition number of the transformation.

---

[3]We overload terminology somewhat, and refer to this as a basis change of the representation itself.

**Proposition 4** (Variation of Loss Along Orbits). *Let $p$ be a homogeneous bivariate polynomial. Let $f(p)$ be the true solution as in equation 1 to the problem in Section 3. Let $\mathcal{N}(p)$ be the equivariant neural network prediction. Then*

$$\epsilon_1 \leq \frac{\|\mathcal{N}(p) - f(p)\|_{\text{Fro}}}{\|f(p)\|_{\text{Fro}}} \leq \epsilon_2 \qquad \Longrightarrow \qquad \frac{\epsilon_1}{\kappa^2} \leq \frac{\|\mathcal{N}(gp) - f(gp)\|_{\text{Fro}}}{\|f(gp)\|_{\text{Fro}}} \leq \kappa^2 \epsilon_2, \qquad (6)$$

*where $\kappa$ is the condition number of $g^{[d]}$.*

*Proof.* We calculate directly

$$
\begin{aligned}
\frac{\|\mathcal{N}(gp) - M(gp)\|}{\|M(gp)\|} &= \frac{\|g^{[d]}(\mathcal{N}(p) - f(p))g^{[d]^T}\|}{\|g^{[d]}f(p)g^{[d]^T}\|} \\
&= \frac{\|(g^{[d]} \otimes g^{[d]})\text{vec}(\mathcal{N}(p) - f(p))\|}{\|(g^{[d]} \otimes g^{[d]})\text{vec}(f(p))\|} \\
&\leq \frac{\sigma_{\max}(g^{[d]})^2\|\text{vec}(\mathcal{N}(p) - f(p))\|}{\sigma_{\min}(g^{[d]})^2\|\text{vec}(f(p))\|} \\
&\leq \kappa^2 \epsilon_2.
\end{aligned}
\qquad (7)
$$

The lower bound is shown by taking the minimum singular value in the numerator and maximum in the denominator. □

