# OpenReview forum: "Learning Polynomial Problems with $SL(2, \mathbb{R})$-Equivariance"
_ICLR.cc/2024/Conference — ICLR 2024 poster_

### Official Review · Reviewer_J1SQ · 2023-10-17

**Soundness:** 4 excellent
**Presentation:** 3 good
**Contribution:** 4 excellent
**Rating:** 8
**Confidence:** 5

**Summary:**

This paper considers the problem of learning how to provide SDP positivity certificates for polynomials. This problem can be solved using convex solvers but this is typically rather time consuming.

The paper observes that the mapping from positive polynomials to their `maximal entropy' SDP solution is SL(d) equivariant. Focusing on the d=2 case,the paper suggests an SL(2) equivariant architectures based on the Clebsch-Gordan methodology often used for SO(3) and other groups. In practice, this architecture does not perform as well as augmentation based on SO(2) equivariant baselines. The paper suggests an interesting theoretical find to (possibly) explain this: While the Clebch-Gordan architecture can construct all equivariant polynomials, the equivariant function considered in the paper cannot be approximated by equivariant polynomials.

**Strengths:**

1. I am not aware of previous work considering the problem the paper considered: learning SDP positivity certificates. Given the high time complexity of these solvers, their centrality in convex programming, and the fact that certificates are verifiable as explained in the paper, I believe this is a very interesting problem to consider and should be considered further. The paper does a good job, in my opinion, of setting up a first empirical and theoretical baseline to consider this problem.

2. Writing is good, it is an interesting story to read.

3. Theorem 1 regarding non-universality seems an interesting result (despite possible error, and needing some tuning down or context as I discuss in the questions part)

**Weaknesses:**

1. I have some issues re the technical details of the main theorem and the premises of the method, see below. If these issues prove to be non-issues I will raise the score
2. The architecture that actually works is rather basic: MLPs with augmentations. On the other hand one could credit the paper in finding the equivariant structure and hence what the relevant augmentations are.
3. The argument that the SL_2(R) equivariant architecture doesn't work because of lack of universality is difficult to actually substantiate. There are many reasons why an architectures may not work well. Maybe a different SL_2 equivariant architecture will work better?

**Questions:**

The formulation of finding the positive-definite witness with maximal determinant assumes that there are many such witnesses. Are there many witnesses? e.g. when we discuss polynomials of degree 2 and the monomail vector is (x,y) I think that a symmetric matrix uniquely Q uniquely defines a quadratic polynomial (x,y)Q(x,y).
When we discuss polynomials of higher degree there are ambiguities that come from the fact that, say, (x^2)(y^2)=(xy)(xy). But this can be dealt with directly by adding more symmetry constraints into the matrix. In other words, the matrix should be a moment matrix as defined in [Lasserre 2001]. Once these constraints are added I believe that there will be no more ambiguities. Do you agree? If so wouldn't it make sense to incorporate the symmetries and forget about optimizing over logdet?

I have two issues with the non-universality proof. The first issue has to do with the correctness of the proof. In the proof of theorem 1 you display the matrix f(x^8+y^8) (let's call it M) which was computed numerically using Mosek. Is this matrix really a factorization of x^8+y^8?
If I understood everything correctly, denoting v=[x^4, x^3y,...,y^4]^T we should have that for all x,y
x^8+y^8=v^TMv
is this correct? Trying this on numpy with the M you specified and x=1, y=1 I get
v^TMv=1.76
while for x=1 y=1
x^8+y^8=2
Note also that the trivial factorization of x^8+y^8 would be M0=diag(1,0,0,0,1). which is not in the domain since det(M0)=0. Thus I would suspect that this polynomial is not in the domain of f. Is that true? Or is it possible for a polynomial to have different factorizations of different ranks? Authors please let me know if there is something I misunderstood of if there is some error. Due to this possible error I'm currently setting the rating at 5 and soundness at 2. I will be happy to raise the rating if there is in fact no error.


A second issue is with the result concerning the non-universality of the SL(2) network is not correctness but just about the exposition. It is neat that you prove that the function f you're actually  interested in cannot be approximated by SL(2) equivariant polynomials. But I do think you should note that your function f is not defined on all of the vector space: namely f(p) is only defined if p is indeed positive, and moreover there exists a *strictly* positive definite matrix verifying this. So f is defined on some subset of your vector space. The universality results in [Bogatskiy] pertain to the complex SL_2, but also to functions continuous on the whole domain, and this may end up being the more substantial difference. Another example: in  [Villar et al.] all continuous functions invariant with respect to the non-compact Lorenz group action are shown to be approximated by polynomials. Here again the continuous functions are defined on all of the domain.

Another angle to think of these issues is: For non-compact groups often distinct orbits cannot be separated by continuous functions. For example: consider the action of SL_d on d by d matrices by multiplication from the right: you can see that a d by d matrix which does not have full rank, say A=diag(0,1,1,...,1), is not in the same orbit as the zero matrix, but its orbit contains all matrices of the form diag(0,epsilon,..,epsilon) and thus any SL_d *invariant* function F continuous on all of the domain will satisfy F(A)=F(0). For more on this see [Dym and Gortler] Section 2.5 and Section 1.4, especially the paragraph titled `algebraic separation vs. orbit separation'.

So to be concrete about this: I think you should mention in the paper that the function f is not defined everywhere, and would suggest to change the paragraph `why is SL(2,R) different' and other places where this issue is discussed, to note that this also might be a reason for the difference between universality results elsewhere and your non-universality result here.

Other remarks, questions, suggestions, according to order in the paper and not importance:
Somewhere in the paper- explain why you decided to restrict yourselves to polynomials of two variables.

In your discussion of Schur's Lemma in page 6: the lemma applies to complex representations and not real. Do you address this (if not, maybe just add a disclaimer)?

Page 4: when you introduce the function f discuss its domain. Mention that in its domain the function is well defined since the opimization problem has a unique maximizer.

Page 6: I didn't understand your explanation of the last layer.

Page 8 timing: The accuracy you achieve is not bad, but probably can be achieved by first order methods which can be much fast than Mosek. You should at least mention this, even if you do not compare against such a method in practice.

Page 9: you reference the wrong paper by Puny. You meant [Puny 2021] not [Puny 2023]





References mentioned above:
[Villar et al.]  Scalars are universal: Equivariant machine learning,
structured like classical physics
[Dym and Gortler] Low Dimensional Invariant Embeddings for Universal Geometric
Learning
[Puny 21]  Frame averaging for invariant and equivariant network design
[Lasserre 2001] Global Optimization with polynomials and the problem of moments.

---

> ### Author Response · Authors · 2023-11-17
> **Response to the reviewer's comments/questions (1/2)**
>
> We sincerely thank the reviewer for their thorough review of our paper, and for acknowledging the significance of the proposed problem and our contribution to it. We now respond to individual points below.
>
> ### Weakness 1: technical detail clarification
>
> We clarify the reviewer’s points in the sections below. Indeed, we believe these issues are non-issues, but the reviewer should please feel free to follow up if anything remains unclear (and we very much thank the reviewer for their attention to detail).
>
> ### Weakness 2: MLP with augmentations performs best
>
> Indeed, the message of our empirical findings is that a simple method (data augmentation) works best, although as noted in Section 5, some care must be taken to augment with reasonably well-conditioned elements of $SL(2,\mathbb{R})$. This could save practitioners time, in not designing a complex polynomial-based equivariant architecture; it could also prompt the development of different equivariant paradigms.
>
> ### Weakness 3: Why our $SL(2,\mathbb{R})$-equivariant architecture failed, and whether another architecture will work better
>
> The reviewer is correct that multiple factors can contribute to the failure of an architecture to train. However, we demonstrate not just that the architecture isn’t universal, but that it can’t represent the very function we would like to train it to represent for positivity verification. Therefore, it is at the very least one problem in our experiments with this architecture. Moreover, the conclusion of Section 4.3 is not just that this particular architecture will not work -- it’s that the function we are trying to learn cannot be approximated by any equivariant polynomial. Therefore, any architecture based on approximation via polynomials will fail for this application. However, we agree that designing an architecture that goes beyond approximating equivariant polynomials is a natural future direction. Should such an architecture continue to underperform relative to an augmented MLP, it would likely then be due to another reason.
>
> ### Question: Certificate degrees of freedom
>
> As you pointed out, for the degree 2 case there is a unique feasible point. However, as soon as the polynomial is degree 4, there is a degree of freedom. For example, consider $p = x^4  - x^2y^2 + y^4$. For any $z$, a matrix of the form $$Q =
> \begin{pmatrix} 1 & 0 &      z\\\  0 &-1-2z& 0\\\ z &0  &     1
> \end{pmatrix}$$
> is a witness for the positivity of $p$. Restricting that the main antidiagonal has equal entries (like in a moment matrix) would give the matrix
> $\begin{pmatrix}1&   0&    -\frac13\\\ 0 &   -\frac13&  0\\\ -\frac13 & 0  &  1\end{pmatrix}$,
> which does not have a positive determinant. On the other hand,
> $\begin{pmatrix}1  &    0  & -.75 \\\ 0  & .5 & 0\\\ -.75 &  0 & 1\end{pmatrix}$
> is also a witness (with $z = -.75$), and the eigenvalues are positive. This is just one example of why additional restrictions will sometimes prohibit you from finding a PSD certificate. (See Example 4.1 of [Parrilo] for another example of how playing with the free parameter may uncover a PSD certificate). We selected the certificate of maximal determinant because it is the analytic center of all such possible witnesses.
>
> To be clear, we do require $(xy)(xy) = x^2  y^2$. For example, consider the following bilinear form:
> $$ \begin{pmatrix} x_1^2 \\\ x_1y_1 \\\ y_1^2 \end{pmatrix}^T \begin{pmatrix}1 & 2& 3\\\ 2 & 4& 5\\\ 3& 5& 6\end{pmatrix}  \begin{pmatrix} x_2^2 \\\ x_2y_2 \\\ y_2^2 \end{pmatrix} =\cdots + 3y_1^2x_2^2 + 4x_1y_1x_2y_2+3x_1^2y_2^2 + \cdots$$
> Because we require $x_1 = x_2$ and $y_1 = y_2$, these 3 terms collapse to $10 x^2y^2$. However, there is no reason to require the coefficient 3 to equal the coefficient 4.
>
> ### Question: Rounding in non-universality proof
> You are correct that the matrix M does not precisely satisfy the linear equations. However, this is due to rounding (which we tried to imply by using “$\approx$”). As you pointed out, this is confusing, so we can include more digits
> $$M =
> \begin{pmatrix}
> 1&  0& -1.56344& 0& 1/3 \\\ 0 &3.12688& 0& -8/3& 0 \\\ -1.56344 &0 &14/3 &0& -1.56344 \\\ 0 &-8/3 &0 &3.12688& 0 \\\ 1/3 &0& -1.56344& 0& 1
> \end{pmatrix} $$
> This exactly represents $x^8 + y^8$ and the determinant is $2.3703703172386135$, which is to several significant figures the determinant of M found from Mosek.
>
> It is possible that different matrices representing a polynomial have different ranks – one example is exactly $x^8 + y^8$, like you point out. Any binary form positive on $\mathbb{R^2}\setminus (0,0)$ is in the domain of $f$ because there exists a feasible full rank matrix. In particular, $x^8 + y^8$ is in the domain of $f$.

---

> > ### Author Response · Authors · 2023-11-17
> > **Response to the reviewer's comments/questions (2/2)**
> >
> > ### Question: Additional explanations for the approximation counterexample
> >
> > The proof of Bogatsky [2] cites the theorems in Yarotsky [3], which are of the form “for every compact subset…”. The theorems only imply that the functions are approximated on compact subsets. Therefore, one could take a compact subset including $x^8 + y^8$, and then this theorem would fail. Hence, we do not think the issue is because the domain of $f$ is not the whole space, but we agree mentioning it is important because it is yet another difference between our case and the existing theorems (and have added this to the draft).
> >
> > ### Question: Impossibility of separating certain orbits for continuous $SL(2,\mathbb{R})$-invariant functions
> >
> > We thank the reviewer for pointing out this interesting fact about continuous, $SL(2,\mathbb{R})$-invariant functions. We view this as an interesting structural fact about what continuous invariant functions exist, independently of how well they may be represented. One can still ask how well such a function can be approximated by polynomials -- since polynomials are continuous too, they will also exhibit this property, in which the function must take the same value on the orbit of the 0 matrix and the orbit of $\text{diag}(0,1,...,1)$. However, our impossibility result involves a particular continuous, equivariant function $f$ that already exists (indeed, it is the exact function that we care about). We then demonstrate that this function cannot be approximated by any equivariant polynomial.
> >
> >
> > ### Responses to remaining remarks/questions/suggestions (in order)
> >
> > * We restricted ourselves to polynomials of two variables for simplicity, but polynomials in more than two variables are a natural next step. We will note this in the paper.
> > * We have mentioned that we need a real version of Schur’s lemma, which is less specific about the form of possible intertwiners.
> > * Page 4:  Thank you for this suggestion; we have added a mention of the domain after the definition.
> > * Page 6: We have modified the description of the last layer, moving the technical details of how $L$ is computed to the appendix. Is it any more clear now?
> > * Page 8: For large problems, we agree that a first order method may be much faster. We have changed the “solve” timings to those from SCS, a first order solver, so a comparison of accuracy of a first order solver can also be made. Some of these times are slightly higher than for Mosek, which we suspect is because these problems are still not large, and the overhead in setting up the SCS problem may be dominating. Nonetheless, these results demonstrate that an MLP is much faster than both Mosek and a first order method.
> > * Page 9: Thank you for pointing out the reference mistake. We have corrected it.
> >
> > ### References
> >
> > [1] Parrilo, Pablo A. Structured semidefinite programs and semialgebraic geometry methods in robustness and optimization. California Institute of Technology, 2000.
> >
> > [2] Alexander Bogatskiy, Brandon Anderson, Jan T Offermann, Marwah Roussi, David W Miller, and Risi Kondor. Lorentz group equivariant neural network for particle physics. In Proceedings of the 37th International Conference on Machine Learning, 2020.
> >
> > [3] Yarotsky, Dmitry. "Universal approximations of invariant maps by neural networks." Constructive Approximation 55.1 (2022): 407-474.

---

> > > ### Comment · Reviewer_J1SQ · 2023-11-19
> > >
> > > Thanks for the insightful comment. I am now pretty convinced in the technical soundness of the paper (though still relying on Mosek for a proof is somewhat problematic). I think it is a very interesting paper overall. I raised my score to 8. Good job!

---

### Official Review · Reviewer_b6w7 · 2023-11-01

**Soundness:** 3 good
**Presentation:** 2 fair
**Contribution:** 3 good
**Rating:** 8
**Confidence:** 3

**Summary:**

This paper proposes a novel approach to learning polynomial problems with -equivariance. The authors demonstrate the effectiveness of neural networks in solving polynomial problems in a data-driven fashion, achieving tenfold speedups while retaining high accuracy. They also adapt their learning pipelines to accommodate the structure of the non-compact group , including data augmentation and new -equivariant architectures. The paper presents a thorough analysis of the proposed approach, including theoretical proofs and experimental results.

**Strengths:**

+The paper presents a novel approach to solve polynomial problems with -equivariance, which is a significant contribution to the field.
+ The authors provide a detailed analysis of the mathematical properties of the proposed approach, including its equivariance and homogeneity properties. This analysis is essential for understanding the theoretical foundations of the approach.
+The authors provide a detailed comparison with existing methods, highlighting the advantages of their approach.

**Weaknesses:**

- The paper could benefit from more detailed explanations of some of the technical concepts and methods used, particularly for readers who are not familiar with the field. For example, the paper could provide more details on the mathematical background of  and its relevance to the problem at hand.

- The paper could provide more details on the implementation of the proposed approach, including the datasets used in the experiments, the choice of neural network architecture and optimization algorithm.

- The paper could benefit from a more detailed discussion of the limitations and potential future directions of the proposed approach.

- While the proposed architecture is effective for learning equivariant polynomials, the LACK OF UNIVERSALITY mentioned could limit its applicability to more complex or diverse datasets. This could be a potential drawback when applying the proposed approach to real-world problems.

- While the experimental results are promising, the authors could provide more detailed analysis and discussion of the results to further support their claims. For example, the paper could provide more details on the sensitivity of the proposed approach to hyperparameters and the robustness of the approach to noisy data.

**Questions:**

Please check the Weaknesses listed above.

---

> ### Author Response · Authors · 2023-11-17
> **Thank you for the review**
>
> We thank the reviewer for their positive review of our paper.

---

### Official Review · Reviewer_AJ4x · 2023-11-03

**Soundness:** 1 poor
**Presentation:** 1 poor
**Contribution:** 2 fair
**Rating:** 5
**Confidence:** 2

**Summary:**

This paper poses to solve certain polynomial optimization problems using architectures which respect the SL(2,R) symmetry.

But most of the critical details are looking very opaque.

**Strengths:**

The paper has definitely identified a very novel use case for neural nets – like positivity certification for polynomials.

The experimental data also seems reasonable.

**Weaknesses:**

What is this $\psi_n$ function in equation 2? This does not look like a Clebsch-Gordon coefficient.

Section 4.2 is extremely vague. The pseudocode is almost unreadable because it is calling functions (in lines 8 and 10) which has never been defined. Also, the entire motivation of this Section seems unclear to me, even if I assume the correctness of Lemma 1. How is this related to the training problem that eventually seems to be the target?

The issues delineated in Section 4.3 do not seem relevant to the immediate question at hand which are all about certain polynomial optimizations. Or am I missing something? It would have been much better to use the space to explain what the experimental setup. Like it seems pretty critical to understand what is the author’s idea of a “natural” polynomial and these details are missing from the main paper! The loss functions used in this experiment also seem to be not clearly specified and that makes it further challenging to understand what is happening.

**Questions:**

Q1.

Why is SL(2,R) equivariance crucial to the usecases identified here?

Its not possible to make the connection between this group and the problem as stated in equation 1.

Q2.

What is the training time for the nets involved in Table 2? I guess what is reported as “MLP times” are just the inference times, right?

But the timings specified for the other methods are probably the “total” time they take to run and there are no other time costs there.

Q3.

What is the full and explicit specification of the loss function that is being optimized in the experiment in Section 5?

And how does this respect SL(2,R)?

---

> ### Author Response · Authors · 2023-11-17
> **Response to the reviewer's comments/questions (1/2)**
>
> We thank the reviewer for providing feedback on our paper, and for highlighting the novelty of the positivity certification application. We hope to clarify the reviewer’s questions below.
>
> ### Transvectant/Clebsch-Gordan Clarification
>
> Although the form may look unfamiliar, the transvectant, which we denote by $\psi$, indeed describes precisely the map from two irreps to their tensor product’s decomposition back into irreps. It may be helpful to recall that the finite dimensional irreducible vector spaces of $SL(2,\mathbb{R})$ can be identified with the homogeneous polynomials of a given degree, so the Clebsch-Gordan coefficients for $SL(2,\mathbb{R})$ should map two input polynomials to an output polynomial. ($\psi_n$ indeed maps two input polynomials to an output polynomial.) Please also see the first paragraph of the introduction to Böhning [1] for formal clarification that the transvectant operation, which we denote by $\psi$, is exactly the Clebsch-Gordan map for this group.
>
> ### Section 4.2 and pseudocode
>
> Thank you for pointing out the ambiguity in the functions in the pseudocode in Section 4.2. We have corrected this in the updated draft.
>
> We have also rewritten the description of the last layer in Section 4.2, deferring the exact details of the mathematical derivation (which may have been confusing) to the appendix.
>
> ### Relevance of Section 4.3 to polynomial optimizations, and experimental details
>
> Positivity verification, as described in Section 3, is an equivariant learning task. Therefore, one might naturally design an equivariant architecture for this problem, as we do in Sections 4.1-4.2. The surprising, and central, theoretical contribution of this paper is that an equivariant network ostensibly designed for the very purpose of positivity verification, is in fact unable to approximate the function of interest for positivity verification. Therefore, any network designed to solve this problem must be able to output more than just equivariant polynomials, which is a significant restriction on the architectural design space relative to prior work! To the best of our knowledge, such a finding is also without precedent in the equivariance literature. It also provides some motivation for using data augmentation, rather than an equivariant architecture. This is the content of Section 4.3. Corollary 1 implies that it doesn’t matter what loss function we use -- no equivariant polynomial can approximate the function of interest under any standard loss function. In the paper, we used the normalized mean squared error loss (with a small additive stabilization term in the denominator), as noted in the caption of Figure 2.
>
> By “natural” polynomials, we just meant that they come from a naturally occurring, application-specific distribution, such that it is empirically possible to well-approximate an NP-hard optimization like polynomial positivity verification over this distribution. (Intuitively, we would expect such a distribution to have low entropy -- it should differ significantly from a uniform distribution over polynomials in a norm-bounded ball, e.g.) For example, one of our experiments is on the polynomials that arise from spherical code bounds; this provides one example of a “natural” set of polynomials, but there is no single definition of such a distribution. This point should not be critical to our main arguments.
>
> ### Q1: Utility of $SL(2,\mathbb{R})$ equivariance
>
> While the symmetry group is not crucial for studying the problem, equivariant learning has been helpful from a sample complexity perspective in many other contexts. Since our positivity verification problem is equivariant with respect to the large group  $SL(2,\mathbb{R})$, it is natural to try exploiting this structure in the learning pipeline. Please see the paragraph after (1) for an explanation of the precise equivariance property, and please feel free to follow up with further questions if this point remains unclear.
>
> ### Q2: Clarification on timings
>
> Indeed, the MLP times are for inference, while the Mosek times are full solves. The motivation of this work is to speed up the *online* determination of nonnegativity, which is possible with just MLP inference. Amortizing the runtime of a traditional solver over many instances is not possible, hence why we compare to the full solve time. For reference, the training time for the MLP was on the order of 90 minutes, which even included many different loss evaluations (e.g. evaluating the loss under random $SL(2,\mathbb{R})$-augmentations).

---

> > ### Author Response · Authors · 2023-11-17
> > **Response to the reviewer's comments/questions (2/2)**
> >
> > ### Q3: Loss function
> >
> > The loss function was normalized mean squared error (as noted in Figure 2), computed as the Frobenius norm of the error between f(p) and the network output (which are matrices of equal size). In other words, if the output matrix is $M$ and the correct matrix $f(p)$ is $N$, the loss function is $\frac{\sum_{i,j}(M_{ij} - N_{ij})^2}{1+\sum_{i,j}N_{ij}^2}$ (where we add $1$ to the denominator for stability), averaged over all data points. We have added the explicit equation to the caption of Figure 2.
> >
> > ### References
> >
> > [1] Böhning, Christian, and Hans-Christian Graf V. Bothmer. "A Clebsch–Gordan formula for SL3 (C) and applications to rationality." Advances in Mathematics 224.1 (2010): 246-259.

---

> > > ### Author Response · Authors · 2023-11-20
> > > **Responding to second part of Q3**
> > >
> > > We realized we had not answered the reviewer’s question on whether the loss function respects $SL(2,\mathbb{R})$, and so clarify it here. The loss function is *not* $SL(2,\mathbb{R})$-invariant, and this is a subtle distinction from the case of compact group equivariance (e.g. many loss functions are invariant under rotations) and a good observation by the reviewer.  We explicitly bound how the loss function varies along an orbit in Proposition 4 of the Appendix, in accordance with the condition number of the element of $SL(2,\mathbb{R})$ (which is related to why we augment only with well-conditioned elements of $SL(2,\mathbb{R})$). Finding an appropriate $SL(2,\mathbb{R})$-invariant loss function is an interesting future question.
> > >
> > > If the reviewer has any remaining questions or concerns, we would be very happy to correspond further during the remainder of the discussion period.

---

### Author Response · Authors · 2023-11-17
**Summary of updates to draft**

We thank all reviewers for their feedback, and respond individually to their points. We have uploaded a revised draft, with the following minor changes:

* We clarified the definitions of functions in the pseudocode in Algorithm 1
* We rewrote the description of the last layer in Section 4.2, deferring the exact details of how we compute $L$ to the appendix.
* We have added the explicit loss equation to the caption of Figure 2.
* We contextualize our impossibility result below Corollary 1, noting that the function is not defined on the whole space.
* We have specified that we need a real version of Schur’s lemma.
* We have added a mention of the domain after equation (1).
* We have changed the “solve” timings to those from SCS, a first order solver, so a comparison of accuracy of a first order solver (which one would expect to be even faster) can also be made. The results demonstrate that the forward pass is much faster than both Mosek and a first order method.
* We corrected the reference to Puny et al 2021.

---

### Meta-Review · Area_Chair_VR8g · 2023-12-14

**Metareview:**

The paper proposes ML approaches to speed up the procedures for certifying positivity of polynomials. They make use of certain symmetries in these problems. They experiment with both data augmentation (based on such symmetries) and using an equivariant architecture.

**Justification For Why Not Higher Score:**

Less clear there is great interest, or much enthusiasm from reviewers.

**Justification For Why Not Lower Score:**

Seems clearly above borderline.

---

### Decision · Program_Chairs · 2024-01-16

Accept (poster)